# A novel SUN1-ALLAN complex coordinates segregation of the bipartite MTOC across the nuclear envelope during rapid closed mitosis in *Plasmodium berghei*

**Mohammad Zeeshan[1†§], Igor Blatov[1†], Ryuji Yanase[1‡], David JP Ferguson[2‡], Sarah L Pashley[1], Zeinab Chahine[3], Yoshiki Yamaryo-Botté[4], Akancha Mishra[1], Baptiste Marche[3], Suhani Bhanvadia[3], Molly Hair[2], Sagar Batra[1], Robert Markus[1], Declan Brady[1], Andrew R Bottrill[5], Sue Vaughan[2], Cyrille Y Botté[4], Karine G Le Roch[3], Anthony A Holder[6], Eelco Tromer[7], Rita Tewari[1*]**

[1]University of Nottingham, School of Life Sciences, Nottingham, United Kingdom; [2]Oxford Brookes University, Department of Biological and Medical Sciences, Oxford, United Kingdom; [3]Department of Molecular, Cell and Systems Biology, University of California, Riverside, Riverside, United States; [4]Apicolipid Team, Institute for Advanced Biosciences, CNRS UMR5309, Université Grenoble Alpes, Grenoble, France; [5]School of Life Sciences, Gibbet Hill Campus, University of Warwick, Coventry, United Kingdom; [6]Malaria Parasitology Laboratory, The Francis Crick Institute, London, United Kingdom; [7]Cell Biochemistry, Groningen Biomolecular Sciences and Biotechnology Institute, Faculty of Science and Engineering, University of Groningen, Groningen, Netherlands

*For correspondence:
rita.tewari@nottingham.ac.uk

†These authors contributed equally to this work
‡These authors also contributed equally to this work

Present address: §Division of Molecular Microbiology and Immunology, CSIR-Central Drug Research Institute, Lucknow, India

Competing interest: The authors declare that no competing interests exist.

## eLife Assessment

In Plasmodium male gametocytes, rapid nuclear division occurs with an intact nuclear envelope, requiring precise coordination between nuclear and cytoplasmic events to ensure proper packaging of each nucleus into a developing gamete. This **valuable** study characterizes two proteins involved in the formation of *Plasmodium berghei* male gametes. By integrating live-cell imaging, ultrastructural expansion microscopy, and proteomics, this study **convincingly** identifies SUN1 and its interaction partner ALLAN as crucial nuclear envelope components in male gametogenesis. A role for SUN1 in membrane dynamics and lipid metabolism is less well supported. The results are of interest for general cell biologists working on unusual mitosis pathways.
[Editors' note: this paper was reviewed by Review Commons.]

**Abstract** Mitosis in eukaryotes involves reorganisation of the nuclear envelope (NE) and microtubule-organising centres (MTOCs). During male gametogenesis in *Plasmodium*, the causative agent of malaria, mitosis is exceptionally rapid and highly divergent. Within 8 min, the haploid male gametocyte genome undergoes three replication cycles (1N to 8N), while maintaining an intact NE. Axonemes assemble in the cytoplasm and connect to a bipartite MTOC-containing nuclear pole (NP) and cytoplasmic basal body, producing eight flagellated gametes. The mechanisms coordinating NE remodelling, MTOC dynamics, and flagellum assembly remain poorly understood. We identify

the SUN1-ALLAN complex as a novel mediator of NE remodelling and bipartite MTOC coordination during *Plasmodium berghei* male gametogenesis. SUN1, a conserved NE protein, localises to dynamic loops and focal points at the nucleoplasmic face of the spindle poles. ALLAN, a divergent allantoicase, has a location like that of SUN1, and these proteins form a unique complex, detected by live-cell imaging, ultrastructural expansion microscopy, and interactomics. Deletion of either SUN1 or ALLAN genes disrupts nuclear MTOC organisation, leading to basal body mis-segregation, defective spindle assembly, and impaired spindle microtubule-kinetochore attachment, but axoneme formation remains intact. Ultrastructural analysis revealed nuclear and cytoplasmic MTOC miscoordination, producing aberrant flagellated gametes lacking nuclear material. These defects block development in the mosquito and parasite transmission, highlighting the essential functions of this complex.

## Introduction

Mitosis, the process of eukaryotic cell division, requires the accurate segregation of nuclear and cytoplasmic materials, coordinated by dynamic changes in the NE and MTOCs (*Liu and Pellman, 2020*). The NE, a double-membrane structure with embedded NP complexes, is a selective barrier, facilitating signal exchange between cytoplasm and nucleus. The NE consists of inner (INM) and outer (ONM) membranes, with the ONM continuous with the endoplasmic reticulum (ER). Beyond structural and transport roles, the NE is integral to mitosis, supporting spindle formation, kinetochore attachment, and chromosome segregation, while also accommodating changes in ploidy (*Dey and Baum, 2021*; *Smoyer and Jaspersen, 2014*).

In most mammalian cells, mitosis is considered 'open,' characterised by complete NE disassembly, chromosome condensation, and segregation via a microtubule-based bipolar spindle. In contrast, in unicellular organisms such as the budding yeast *Saccharomyces cerevisiae*, mitosis is comparatively 'closed:' the NE remains intact, and chromosomes are segregated by an intranuclear spindle anchored to acentriolar NPs (*Boettcher and Barral, 2013*; *Sazer et al., 2014*). However, it is now believed that no mitosis is wholly 'open' or 'closed' and that in most cases, remnants of the NE interact with the

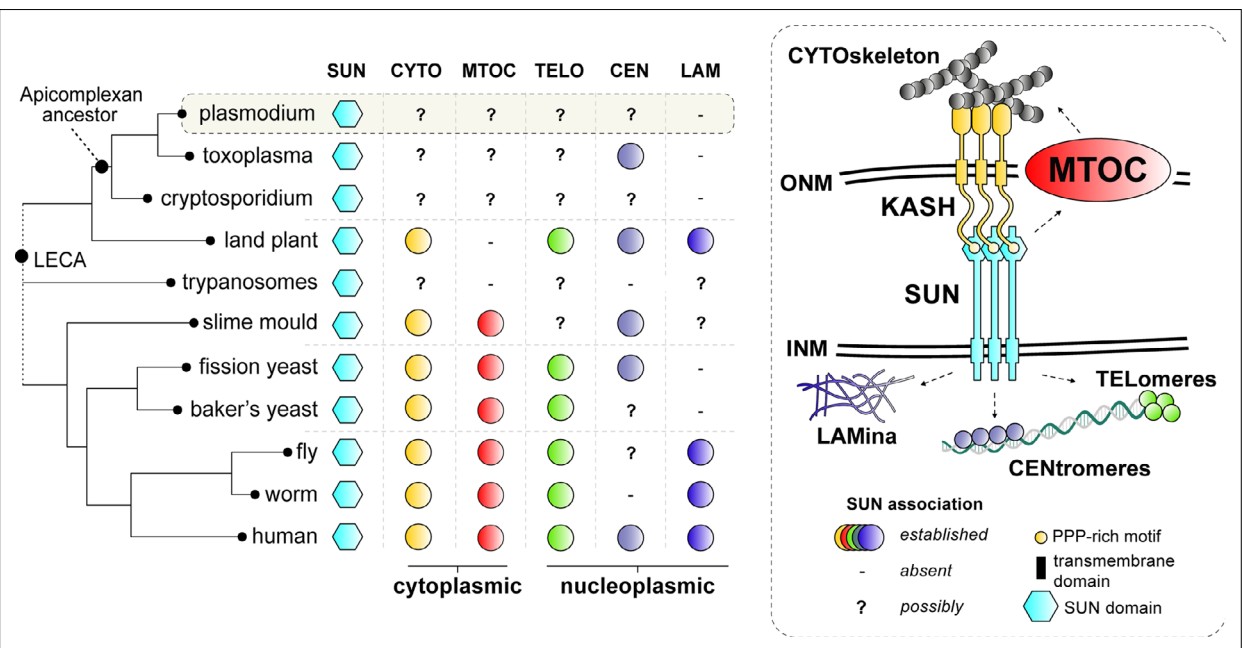

**Figure 1.** Comparative analysis of SUN protein functions in common eukaryotic model systems. SUN proteins bridge the outer (ONM) and inner (INM) membranes of the nuclear envelope (NE) to link the cytoskeleton (i.e. actin, microtubules and its organising centres) to various heterochromatic domains (i.e. centromere, telomeres) and the nuclear lamina (five different functions with a unique colour). Established roles of SUN proteins in common model organisms are depicted by coloured circles. '-' no functional connection for SUN was found and/or structures are not present. '?' denotes possible roles for SUN proteins as NE connections with such structures have been established in these lineages. LECA: Last Eukaryotic Common Ancestor.

spindle to support chromosome segregation (*Dey and Baum, 2021*). Closed mitosis involves diverse NE dynamics: for example, *Trypanosoma* and *Saccharomyces* employ intranuclear spindle assembly, whereas *Chlamydomonas* and *Giardia* assemble spindles outside the nucleus; the NE remains largely unbroken, but large polar fenestrae are formed to allow microtubule access to the chromosomes (*Makarova and Oliferenko, 2016*). These examples highlight the fact that 'open' and 'closed' mitosis are merely two extremes of a continuum of NE remodelling strategies during mitosis, underlining extensive diversity in cell division, a core cellular process.

The interaction between nucleus, the chromatin, and in general the nucleoplasmic environment and the cytoskeleton is facilitated by the LINC (Linker of Nucleoskeleton and Cytoskeleton) complex (*Figure 1*), which bridges the INM and ONM in most eukaryotes via interactions between SUN (Sad1, UNC84)-domain proteins on the INM and KASH (Klarsicht, ANC-1, Syne Homology)-domain proteins on the ONM (*Hao and Starr, 2019*; *Starr and Fridolfsson, 2010*). The LINC complex links chromatin and the cytoskeleton throughout eukaryotes (*Figure 1*). For instance, LINC complexes anchor telomeres to the NE during meiosis in budding yeast (*Schober et al., 2008*) and cluster centromeres near the INM in fission yeast (*Funabiki et al., 1993*). Furthermore, SUN proteins connect to the lamin-like proteins that cover the INM on the nucleoplasmic side in animals and plants (*Koreny and Field, 2016*). In plants, SUN–WIP complexes (analogous to the SUN-KASH proteins of animals and fungi) connect the nucleus to actin filaments (*Zhou et al., 2015*), while *Dictyostelium* uses SUN1 to anchor the spindle pole body-like MTOC at the NE (*Xiong et al., 2008*). Such interactions coordinate nuclear positioning, chromosome organisation, and mechanical integration with the cytoskeleton. Evolutionary studies suggest that LINC was established before the emergence of the last eukaryotic common ancestor (LECA), highlighting its ancient role in NE formation and cytoskeletal coordination (*Figure 1*; *Baum and Baum, 2014*; *Koreny and Field, 2016*). However, as SUN proteins are broadly conserved in many eukaryotic lineages, KASH and lamin-like proteins are often not detected (*Koreny and Field, 2016*). Intriguingly, in many non-model eukaryotes like for instance apicomplexan parasites (e.g. *Plasmodium* and *Toxoplasma*) microscopic observations reveal chromatin structures such as centromeres, telomeres, and the spindle pole body-like MTOCs are to be closely associated with and/or embedded in the NE (*Figure 1*). Have KASH/lamins proteins been lost or have novel systems evolved to support a SUN protein-based functions at the nuclear envelope?

The malaria-causing parasite *Plasmodium spp.* uses closed mitosis with some highly divergent features across its complex life cycle. In the asexual stage in the blood of its vertebrate host, during schizogony there is asynchronous nuclear division without coincident cytokinesis, and during male gametogenesis in the mosquito host there is a unique and rapid form of closed mitosis essential for parasite transmission (*Guttery et al., 2022*; *Sinden et al., 1978*). Following activation in the mosquito gut, the haploid male gametocyte undergoes three rounds of genome replication (from 1N to 8N) within just 6–8 min, while maintaining an intact NE. Concurrently, axonemes form in the cytoplasm, emanating from bipartite MTOCs that consist of intranuclear spindle poles and cytoplasmic basal bodies (BB). This rapid karyokinesis and subsequent cytokinesis results in the production of eight flagellated haploid male gametes within 15 min (*Guttery et al., 2022*).

The speed of male gametogenesis in *Plasmodium* imposes unique requirements and constraints on cellular structures. Notably, the flagella lack intraflagellar transport (IFT), which is atypical (*Sinden et al., 2010*). The bipartite organisation of the MTOC was recently revealed, and the use of fluorescently tagged markers such as SAS4 and kinesin-8B has illuminated the dynamics of BB and axoneme formation (*Zeeshan et al., 2022a*; *Zeeshan et al., 2019a*). Kinetochore proteins like NDC80 display an unconventional, largely clustered linear organisation on the spindle, redistributing only during successive spindle duplication (*Zeeshan et al., 2020*). Remarkably, successive spindle and BB segregation occurs within 8 min of gametocyte activation, without NE breakdown, indicating an unusual, streamlined closed mitotic process (*Zeeshan et al., 2022a*; *Zeeshan et al., 2019a*).

The mechanisms that coordinate the formation and function of the intranuclear spindle poles, the cytoplasmic BB, and the NE remain unclear. Specifically, how the NE is remodelled to accommodate rapid genome replication, and how it affects the organisation and function of the bipartite MTOC remain open questions. We addressed these questions by investigating NE remodelling and MTOC coordination in *Plasmodium berghei* (Pb), using the conserved NE protein SUN1. By combining live-cell imaging, ultrastructural expansion microscopy (U-ExM), and proteomic analysis, we identify SUN1 as a key NE component. Using different mitotic and MTOC/BB markers, we investigate their

coordination and the flexibility of NE during rapid mitosis. Our findings reveal that SUN1 interacts with a novel allantoicase-like protein (termed ALLAN), to form a divergent LINC-like complex without KASH proteins. Functional disruption of either SUN1 or ALLAN impairs BB segregation, disrupts spindle-kinetochore attachment, and results in defective flagellum assembly. Our results highlight a unique divergence of NE remodelling and MTOC organisation in *Plasmodium* gametogenesis from conventional eukaryotic model system found in animals, fungi, and plants.

## Results

### Generation and validation of PbSUN1 transgenic lines

To investigate the role of PbSUN1 (PBANKA_1430900) during *P. berghei* male gametogenesis, we generated transgenic parasite lines expressing a C-terminal GFP-tagged SUN1 protein (SUN1-GFP). The GFP-tagging construct was integrated into the 3′ end of the endogenous *sun1* locus via single-crossover recombination (*Figure 2—figure supplement 1A*), and correct integration was confirmed by PCR analysis using diagnostic primers (*Figure 2—figure supplement 1B*). Western blot analysis detected SUN1-GFP at the expected size (~130 kDa) in gametocyte lysates (*Figure 2—figure supplement 1C*). The SUN1-GFP line grew normally and progressed through the life cycle, indicating that tagging did not disrupt PbSUN1 function. The SUN1-GFP line was used to study the subcellular localisation of PbSUN1 across multiple life cycle stages, and its interaction with other proteins during gametogenesis.

### Spatiotemporal dynamics of SUN1 during male gametogenesis

SUN1 protein expression was undetectable by live-cell imaging during asexual erythrocytic stages of the parasite, but robust expression was visible in both male and female gametocytes following activation in ookinete medium (*Figure 2—figure supplement 1D, E*). In activated male gametocytes undergoing mitotic division, SUN1-GFP had a dynamic localisation around the nuclear DNA (Hoechst-stained), associated with loops and folds formed in the NE during the transition from 1N to 8N ploidy (*Figure 2A* and *Video 1*). The NE loops extended beyond the Hoechst-stained DNA, suggesting an abundant non-spherical NE membrane (*Figure 2A*). To further characterise NE morphology during male gametogenesis, serial block-face scanning electron microscopy (SBF-SEM) was used. Analysis of wild-type male gametocytes revealed a non-spherical, contorted nucleus (*Figure 2B*, *Figure 2—figure supplement 1F*). Thin loops of NE (indicated with arrows) were prominent, consistent with the dynamic and irregular location of SUN1-GFP detected by live-cell imaging. 3D-modelling of gametocyte nuclei further confirmed the irregular, non-spherical structure of the NE (*Figure 2C* and *Figure 2—figure supplement 1G*), which expanded rapidly during the period of genome replication. These findings suggest that the male gametocyte NE is a highly dynamic structure that can expand rapidly to accommodate the increased DNA due to replication during gametogenesis.

### SUN1 dynamics by high-resolution imaging

To resolve the SUN1-GFP location with higher precision, structured illumination microscopy (SIM) was performed on SUN1-GFP gametocytes fixed at 6–8 min post-activation. Fixation of gametocytes caused SUN1-GFP signal coalescence into regions within the NE (*Figure 2—figure supplement 2A*), but an uneven intensity of SUN1-GFP signal around the DNA was observed, with areas of higher intensity (*Figure 2D and E* and *Figure 2—figure supplement 2B*, white arrows) likely representing collapsed forms of the SUN1-GFP-labelled loops observed by live imaging (*Figure 2-figure supplement 2A*). To refine the spatial relationship of SUN1 to the BB and the spindle, ultrastructure expansion microscopy (U-ExM) was used. At 6–8 min post-activation SUN1-GFP was prominently located around the DNA and at the junction of the nuclear MTOC and the BB (*Figure 2F* and *Figure 2—figure supplement 3A, B*). Spindle and axoneme staining with α-tubulin antibodies revealed that some SUN1-GFP foci were located next to spindle poles (*Figure 2G* and *Figure 2—figure supplement 3C*), suggesting a role in MTOC organisation during mitosis. WT-ANKA gametocytes (with no GFP present) did not react with the anti-GFP antibodies (*Figure 2-figure supplement 3D*), confirming the specificity of the GFP signal in SUN1-GFP gametocytes.

### The location of SUN1 relative to key mitotic markers during male gametogenesis

To investigate the association of SUN1 with MTOCs, mitotic spindles, and BB/axonemes during male gametogenesis, its location was compared in real time with that of three markers: the cytoplasmic

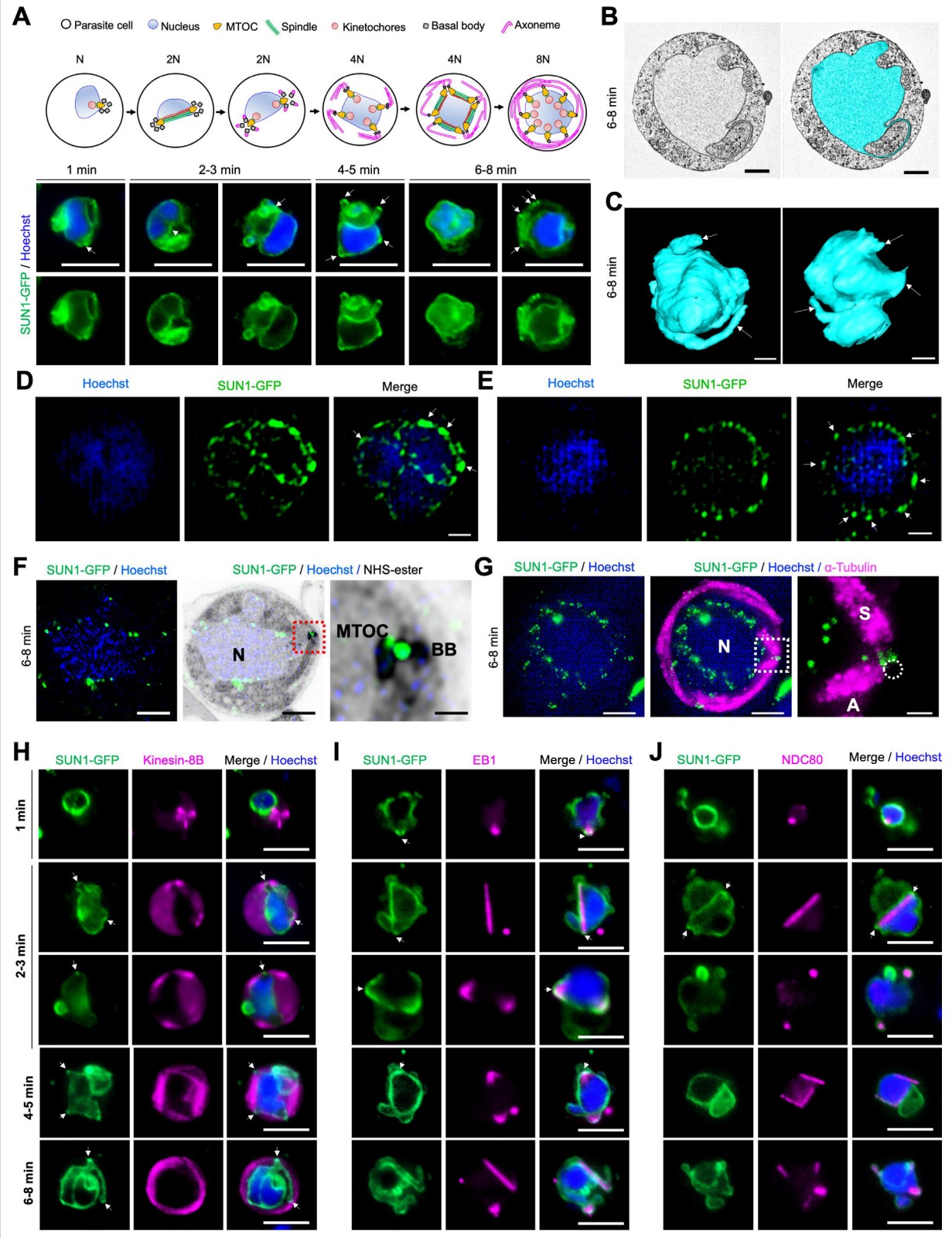

**Figure 2.** Location of SUN1 during male gametogenesis. (**A**) The upper panel schematic illustrates the process of male gametogenesis. N, genome ploidy. Live cell images show the location of SUN1-GFP (green) at different time points (1–8 min) during male gametogenesis. DNA (blue) was stained with Hoechst. White arrows indicate the loop/folds. Representative images of more than 50 cells with more than three biological replicates. Scale bar: 5 μm. (**B**) Serial block face-scanning electron microscopy (SBF-SEM) data slice of a gametocyte highlighting the complex morphology of the

*Figure 2 continued on next page*

*Figure 2 continued*

nucleus (cyan). Representative of more than 10 cells. Scale bar: 1 μm. (**C**) Two 3D models of gametocyte nuclei showing their contorted and irregular morphology. Representative of more than 10 cells. Scale bar: 1 μm. (**D**) SIM images of SUN1-GFP male gametocytes activated for 8 min and fixed with paraformaldehyde. Arrows indicate the SUN1-GFP signals with high intensity after fixation. Representative image of more than 10 cells from more than two biological replicates. Scale: 1 μm. (**E**) SIM images of SUN1-GFP male gametocytes activated for 8 min and fixed with methanol. Arrows indicate the SUN1-GFP signals with high intensity after fixation. Representative image of more than 10 cells from more than two biological replicates. Scale bar: 1 μm. (**F**) Expansion microscopy (ExM) images showing location of SUN1 (green) detected with anti-GFP antibody and BB/MTOC stained with NHS ester (grey). Hoechst was used to stain DNA. Scale bar: 5 μm. Inset is the area marked with the red box around the BB/MTOC highlighted by NHS-ester staining. Scale bar: 1 μm. Representative images of more than 10 cells from two biological replicates. (**G**) ExM images showing location of SUN1 (green) and α- tubulin (magenta) detected with anti-GFP and anti-tubulin antibodies, respectively. Hoechst was used to stain DNA (blue). N=Nucleus; S=Spindle; A=Axoneme. Scale bar: 5 μm. Inset is the area marked with the white box on *Figure 1E* middle panel around the BB/MTOC. Scale bar: 1 μm. Representative images of more than 10 cells from two biological replicates. (**H**) Live cell imaging showing location of SUN1-GFP (green) in relation to the BB and axoneme marker, kinesin-8B-mCherry (magenta) at different time points (1–5 min) during gametogenesis. Blue in merged image is DNA stained with Hoechst. Representative images of more than 20 cells from more than three biological replicates. White arrows indicate the loops/ folds labelled with SUN1 where BB/axonemes are assembled outside the nuclear membrane. Scale bar: 5 μm. (**I**) Live cell imaging showing location of SUN1-GFP (green) in relation to the spindle marker, EB1-mCherry (magenta) at different time points during gametogenesis. Blue in merged image is DNA stained with Hoechst. White arrows indicate the loops/folds labelled with SUN1. Representative images of more than 20 cells from more than three biological replicates. Scale bar: 5 μm. (**J**) Live cell imaging showing location of SUN1-GFP (green) in relation to the kinetochore marker, NDC80-mCherry (magenta) at different time points during gametogenesis. Blue in merged image is DNA stained with Hoechst. White arrows indicate the loops/folds labelled with SUN1. Representative images of more than 20 cells with more than three biological replicates. Scale bar: 5 μm.

The online version of this article includes the following source data and figure supplement(s) for figure 2:

**Figure supplement 1.** Generation of PbSUN1-GFP/ parasites and analysis of subcellular location of SUN1-GFP during blood schizogony and gametogenesis.

**Figure supplement 1—source data 1.** Tiff file of the original gel for *Figure 2—figure supplement 1B*, indicating the relevant band.

**Figure supplement 1—source data 2.** Tiff file of the original gel for *Figure 2—figure supplement 1B*.

**Figure supplement 1—source data 3.** Tiff file of the original gel for *Figure 2—figure supplement 1C*, indicating the relevant band.

**Figure supplement 1—source data 4.** Tiff file of the original gel for *Figure 2—figure supplement 1C*.

**Figure supplement 2.** Localisation of SUN1-GFP in live and fixed male gametocytes.

**Figure supplement 3.** Expansion microscopy (ExM) reveals the SUN1-GFP location along with nuclear envelope (NE) and showing puncta at nuclear microtubule-organising centre (MTOC).

**Figure supplement 4.** Localisation of SUN1-GFP during various stages of parasite development.

BB/axonemal protein kinesin-8B, the spindle microtubule-binding protein EB1, and the kinetochore marker NDC80 (*Zeeshan et al., 2019a*; *Zeeshan et al., 2020*; *Zeeshan et al., 2023*). A parasite line expressing SUN1-GFP (green) was crossed with lines expressing mCherry (red)-tagged kinesin-8B, EB1, or NDC80, and progeny were analysed by live-cell imaging to determine the proteins' spatiotemporal relationships.

Within 1 min post-activation, kinesin-8B appeared as a tetrad marking the BB in the cytoplasm and close to the nucleus, while SUN1-GFP localised to the nuclear membrane, with no overlap between the signals (*Figure 2H*). As gametogenesis progressed, nuclear elongation was observed, with SUN1-GFP delineating the nuclear boundary. Concurrently, the BB (the kinesin-8B-marked tetrad) duplicated and separated to opposite poles in the cytoplasm, outside the SUN1-GFP-defined nuclear boundary (*Figure 2H*). By 2–3 min post-activation, axoneme formation commenced at the BB, while SUN1-GFP marked the NE with small puncta located near the nuclear MTOCs situated between the BB tetrads (*Figure 2H*).

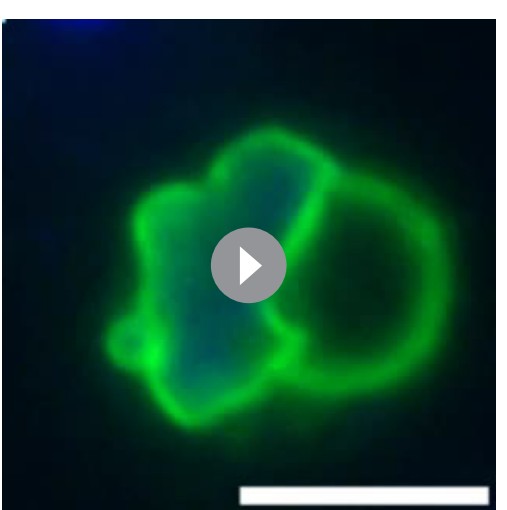

**Video 1.** Time-lapse video showing dynamic location SUN1-GFP in gametocytes activated for 3–4 min. Scale: 5 μm.

https://elifesciences.org/articles/106537/figures#video1

Analysis of gametogenesis with SUN1-GFP and the spindle microtubule marker EB1 revealed a localised EB1-mCherry signal near the DNA, inside the nuclear membrane (marked by SUN1-GFP), within the first min post-activation (*Figure 2I*). Loops and folds in the nuclear membrane, marked by SUN1-GFP, partially overlapped with the EB1 focal point associated with the spindle pole/MTOC (*Figure 2I*, average Pearson Correlation Coefficient, $R<0.6$). At 2–3 min post-activation, EB1 fluorescence extended to form a bridge-like spindle structure within the nucleus, flanked by two strong focal points that partially overlapped with SUN1-GFP fluorescence (*Figure 2I*). A similar dynamic location was observed for another spindle protein, ARK2 (*Figure 2—figure supplement 4A*).

Kinetochore marker NDC80 showed a similar pattern to that of EB1. Within 1 min post-activation, NDC80 was detected as a focal point inside the nuclear membrane, later extending to form a bridge-like structure that split into two halves within 2–3 min, and with no overlap with SUN1-GFP (*Figure 2J*). As with EB1, the nuclear membrane loops formed around NDC80 focal points, maintaining close proximity to the kinetochore bridge (*Figure 2J*).

Together, these observations suggest that although SUN1-GFP fluorescence is partially colocated with spindle poles revealed by EB1 fluorescence, there is no overlap with kinetochores (NDC80) or BB/axonemes (kinesin-8B).

## SUN1 location during female gametogenesis and zygote to ookinete transformation

The location of SUN1-GFP during female gametogenesis was examined using real-time live-cell imaging. Within 1 min post-activation, SUN1-GFP was observed to form a half-circle around the nucleus, eventually encompassing the entire nucleus after 6 or 8 min (*Figure 2—figure supplement 4B*). In contrast to SUN1-GFP in male gametocytes, SUN1-GFP in female gametocytes was more uniformly distributed around the nucleus without apparent loops or folds in the NE (*Figure 2—figure supplement 4B*). This pattern probably reflects the absence of DNA replication and mitosis in female gametogenesis, so that the nucleus remains compact.

During zygote to ookinete differentiation and oocyst development, in zygotes SUN1-GFP had a spherical distribution around Hoechst-stained nuclear DNA at 2 hr post-fertilisation (*Figure 2—figure supplement 4C*), but by 24 hr post-fertilisation, as the zygote developed into a banana-shaped ookinete, the SUN1-GFP-marked NE became elongated or oval, possibly due to spatial constraints within the cytoplasm (*Figure 2—figure supplement 4C*).

## SUN1 is essential for basal body segregation and axoneme-nucleus coordination during male gametogenesis

The role of SUN1 was assessed by deleting its gene using a double crossover homologous recombination strategy in a parasite line constitutively expressing GFP (WT-GFP) (*Janse et al., 2006*; *Figure 3—figure supplement 1A*). WT-GFP parasites express GFP at all stages of the life cycle, facilitating phenotypic comparisons. Diagnostic PCR confirmed the correct integration of the targeting construct at the *sun1* locus (*Figure 3—figure supplement 1B*), and this was verified by qRT-PCR analysis, which showed complete deletion of the *sun1* gene in the resulting transgenic parasite (*Δsun1*) (*Figure 3—figure supplement 1C*).

Phenotypic analysis of the *Δsun1* line was conducted across the life cycle, in comparison to a WT-GFP control. Two independent knockout clones (Clone-1 and Clone-4) were examined: the clones exhibited similar phenotypes, and one, Clone-4 was used for further experiments. *Δsun1* parasites produced a comparable number of gametocytes to WT-GFP parasites, but had reduced male gametogenesis, as evidenced by a significant decrease in gamete formation (exflagellation) (*Figure 3A*). The differentiation of zygotes into ookinetes was also reduced (*Figure 3B*).

To evaluate sporogonic development, mosquitoes were fed with *Δsun1* parasites, and oocyst formation was examined. The number of oocysts was markedly reduced in *Δsun1* parasite-infected mosquitoes compared to WT-GFP parasites at days 7, 14, and 21 post-feeding (*Figure 3C*). At day 7, oocysts were comparable in size but failed to grow further, and they had degenerated by day 21 (*Figure 3D and E*) with no evidence of sporozoite production in midgut or salivary glands (*Figure 3F and G*). Transmission experiments revealed that mosquitoes infected with *Δsun1* parasites were unable to transmit the parasite to naïve mice; in contrast, mosquitoes successfully transmitted the WT-GFP parasite, resulting in a blood-stage infection detected 4 d later (*Figure 3H*).

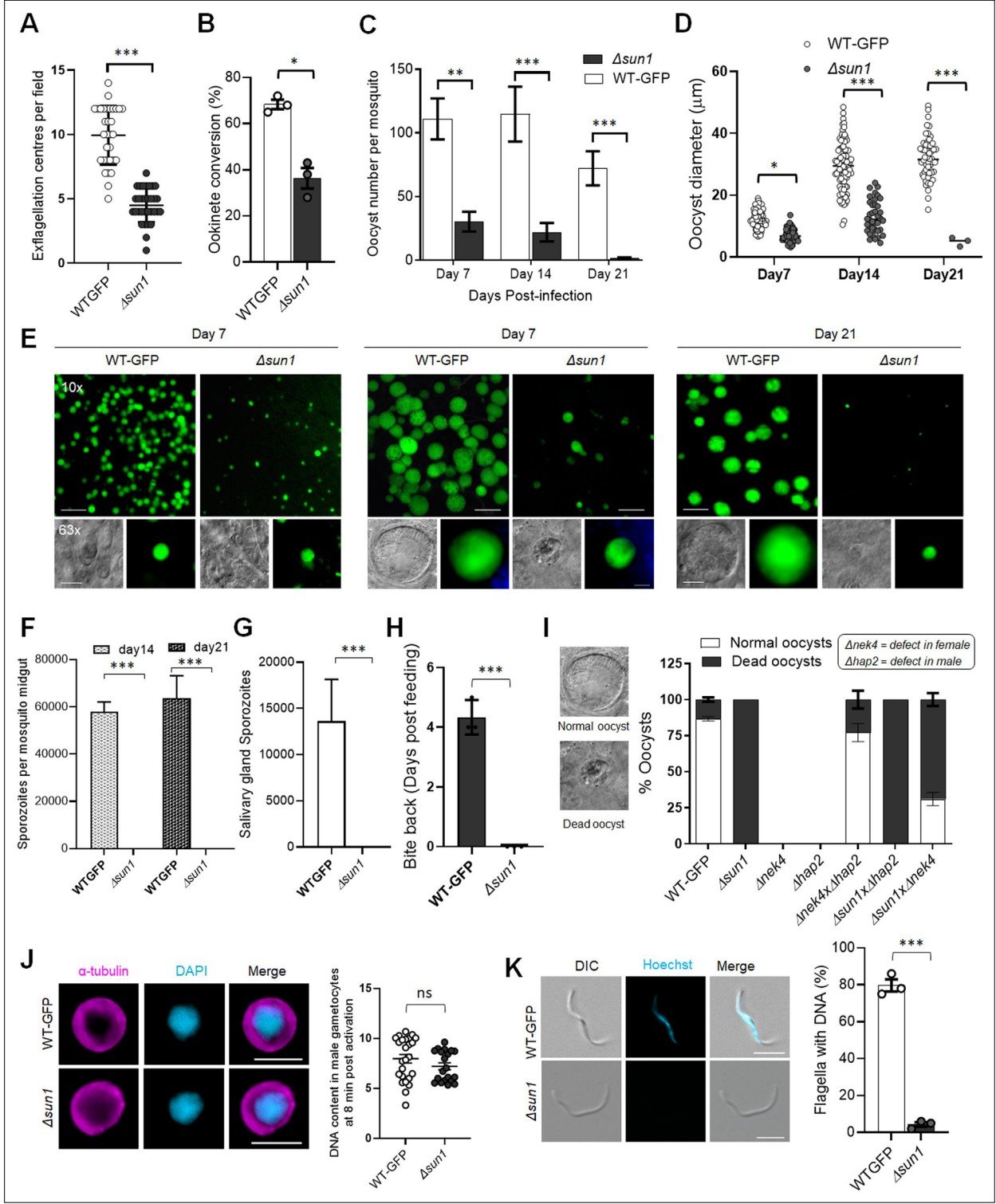

**Figure 3.** Deletion of *sun1* affects male gamete formation and blocks parasite transmission. (**A**) Exflagellation centres per field at 15 min post-activation. n=3 independent experiments (>10 fields per experiment). Error bar ± SEM. (**B**) Percentage ookinete conversion from zygote. n=3 independent experiments (>100 cells). Error bar ± SEM. (**C**) Total number of GFP-positive oocysts per infected mosquito in Δ*sun1* compared to WT-GFP parasites at 7-, 14-, and 21 d post-infection. Mean ± SEM. n=3 independent experiments. (**D**) The diameter of GFP-positive oocysts in Δ*sun1* compared to WT-GFP parasites at 7-, 14-, and 21 d post-infection. Mean ± SEM. n=3 independent experiments. The same WT-GFP data was used to analyse the Δ*allan* mentioned in **Figure 7—figure supplement 1D**. (**E**) Mid guts at 10 x and 63 x magnification showing oocysts of Δ*sun1* and WT-GFP lines at 7-, 14-, and 21 d post-infection. Scale bar: 50 μm in 10 x and 20 μm in 63 x. (**F**) Total number of midguts sporozoites per infected mosquito in Δ*sun1* compared

*Figure 3 continued on next page*

*Figure 3 continued*

to WT-GFP parasites at 14- and 21 d post-infection. Mean ± SEM. n=3 independent experiments. (**G**) Total number of salivary gland sporozoites per infected mosquito in Δ*sun1* compared to WT-GFP parasites at 21 d post-infection. Mean ± SEM. n=3 independent experiments. (**H**) Bite back experiments showing no transmission of Δ*sun1,* while WT-GFP parasites show successful transmission from mosquito to mouse. Mean ± SEM. n=3 independent experiments. (**I**) Rescue experiment showing Δ*sun1* phenotype is due to defect in male *sun1* allele. Mean ± SEM. n=3 independent experiments. (**J**) Representative images of male gametocytes at 8 min post activation stained with DAPI and tubulin (left). Fluorometric analyses of DNA content (**N**) after DAPI nuclear staining (right). The mean DNA content (and SEM) of >30 nuclei per sample are shown. Values are expressed relative to the average fluorescence intensity of 10 haploid ring-stage parasites from the same slide. The same WT-GFP data was used to analyse the Δ*allan* mentioned in *Figure 7—figure supplement 1F*. (**K**) Representative images of flagellum (male gamete) stained with Hoechst for DNA (left). The presence or absence of Hoechst fluorescence was scored in at least 30 microgametes per replicate. Mean ± SEM. n=3 independent experiments. The same WT-GFP data was used to analyse the Δ*allan* mentioned in *Figure 7—figure supplement 1G*. Student's t-test and/or a two-way ANOVA test were employed to assess differences between control and experimental groups. Statistical significance is indicated as *P < 0.05, **P < 0.01, ***P < 0.001, or ns for not significant.

The online version of this article includes the following source data and figure supplement(s) for figure 3:

**Figure supplement 1.** Generation and genotype analysis of *sun1*-knockout (Δ*sun1*) parasites.

**Figure supplement 1—source data 1.** Tiff file of the original gel for *Figure 2—figure supplement 1B*, indicating the relevant band.

**Figure supplement 1—source data 2.** Tiff file of the original gel for *Figure 2—figure supplement 1B*.

**Figure supplement 1—source data 3.** List of genes differentially expressed between Δ*sun1 vs* WT-GFP gametocytes activated for 8 min.

**Figure supplement 2.** Lipidomic analysis of *sun1*-knockout (Δ*sun1*) and WT-GFP parasites.

Because the defect in Δ*sun1* parasites led to a transmission block, we investigated whether the defect was rescued by restoring *sun1* into Δ*sun1* parasites using either the Δ*nek4* parasite that produces normal male gametocytes but is deficient in production of female gametocytes (*Reininger et al., 2005*) or the Δ*hap2* parasite that produces normal female gametocytes but is deficient in production of male gametocytes (*Liu et al., 2008*). We performed a genetic cross between Δ*sun1* parasites and the other mutants deficient in the production of either male (Δ*hap2*) or female (Δ*nek4*) gametocytes. Crosses between Δ*sun1 and Δnek4* mutants produced some normal-sized oocysts that were able to sporulate, showing a partial rescue of the Δ*sun1* phenotype (*Figure 3I*). In contrast, crosses between Δ*sun1* and Δ*hap2* did not rescue the Δ*sun1* phenotype. As controls, Δ*sun1*, Δ*hap2*, and Δ*nek4* parasites alone were examined and no oocysts/sporozoites were detected (*Figure 3I*). To further confirm the viability of controls, male (Δ*hap2*), and female (Δ*nek4*) mutants were crossed together and produced normal-sized oocysts that were able to sporulate (*Figure 3I*). These results indicate that a functional *sun1* gene is required from a male gamete for subsequent oocyst development. To assess the effect of the *sun1* deletion on DNA replication during male gametogenesis, we analysed the DNA content (N) of Δ*sun1* and WT-GFP male gametocytes by fluorometric analyses after DAPI staining. We observed that Δ*sun1* male gametocytes were octaploid (8 N) at 8 min post-activation, similar to WT-GFP parasites (*Figure 3I*), indicating that the absence of SUN1 had no effect on DNA replication. We also checked for the presence of DNA in gametes stained with Hoechst (a DNA dye) and found that most Δ*sun1* gametes were anucleate (*Figure 3J*).

## Transcriptomic and lipidomic analysis of Δ*sun1* gametocytes shows no major changes in overall gene expression and lipid metabolism

To investigate the effect of *sun1* deletion on transcription, we performed RNA-seq analysis in triplicate on Δ*sun1* gametocytes and in duplicate on WT-GFP gametocytes, at 8 min post-activation. Total RNA was extracted to detect changes in gene expression post-activation. A detailed analysis of the mapped reads also confirmed the deletion of *sun1* in the Δ*sun1* strain (*Figure 3—figure supplement 1D*). A relatively small number of differentially expressed genes was identified (*Figure 3—figure supplement 1E*; *Figure 3—figure supplement 1—source data 1*). Gene ontology (GO)-based enrichment analysis of these genes showed that several upregulated genes coded proteins involved in either lipid metabolic processes, microtubule cytoskeleton organisation, or microtubule-based movement (*Figure 3—figure supplement 1F*).

Given the nuclear membrane location of SUN1, we investigated the lipid profile of Δ*sun1* and WT-GFP gametocytes using mass spectrometry-based lipidomic analyses. Lipids are key elements of

membrane biogenesis, with most phospholipids derived from fatty acid precursors (*Figure 3—figure supplement 2A*).

We compared the lipid profile of *Δsun1* and WT-GFP gametocytes, either non-activated (0 min) or activated for 8 min. Both the disruption of *sun1* and the activation of gametocytes caused minimal changes in the amounts of phosphatidic acid (PA), CE, monoacylglycerol (MAG), and TAG. Activated gametocytes had significantly increased levels of PA, CE, and TAG compared to non-activated gametocytes (*Figure 3—figure supplement 2B, C*). Following activation, there were several significant changes in specific molecular species in several lipid classes where the *Δsun1* mutant behaved the opposite way to activated wt. Notably, LPC(16:0), TAG(52:3), TAG(52:4), TAG(52:5), CE(22:4) and CE(22:5), CE(20:3), behaved the opposite way to the WT-GFP (*Figure 3—figure supplement 2D, E*). Though, the disruption of Sun1 has (mild) significant impacts on lipid homeostasis, and response to stimulus for gametocytogenesis, more specifically at the FA compositions of specific molecular classes (precursors and neutral lipid classes), we cannot really conclude on whether this impacts bulk membrane biogenesis and/or lipid signalling but there are significant changes in lipid homeostasis upon Sun1 disruption.

## Ultrastructural microscopy reveals defects in spindle formation, basal body segregation, and nuclear attachment to axonemes in *Δsun1* Gametocytes

To define the ultrastructural defects caused by *sun1* deletion, we used ultrastructure expansion microscopy (U-ExM) and transmission electron microscopy (TEM) to examine the morphology of male gametocytes at critical time points - 1.5-, 8-, and 15 min post-activation, representing the timings of major transitions in spindle formation, BB segregation, and axoneme elongation (*Figure 4A–D* and *Figure 4—figure supplements 1 and 2*).

U-ExM revealed that in WT-GFP gametocytes, spindle microtubules were robustly assembled by 1.5 min post-activation, extending from nuclear MTOCs (spindle poles) to kinetochores. Simultaneously, BB - marked by NHS ester staining - segregated into tetrads, distributed across the cytoplasmic periphery (*Figure 4A*, *Figure 4—figure supplement 1A*). By 8 min, spindles were fully extended, ensuring accurate chromosome segregation, while BB had replicated to nucleate parallel axonemes aligned around the nucleus (*Figure 4—figure supplement 1B*). By 15 min, these processes had culminated in exflagellation, producing mature microgametes containing nuclei tightly associated with BB-derived axonemes (*Figure 4B*, *Figure 4—figure supplement 1C*). In *Δsun1* gametocytes, these processes were severely disrupted. At 1.5 min, α-tubulin staining showed incomplete or malformed spindles. NHS ester staining revealed BB clumped near one side of the nucleus, indicating a failure in segregation (*Figure 4A*, *Figure 4—figure supplement 1A*). By 8 min, spindles remained rudimentary, and BB segregation was still incomplete. Axoneme elongation proceeded, but BB and nuclear poles failed to align, leading to misconnected or unconnected axonemes (*Figure 4—figure supplement 1B*). At 15 min post-activation, the gametocytes show fully assembled axonemes before exflagellation while BBs remain unsegregated (*Figure 4B*).

We performed TEM analysis of *Δsun1* and WT-GFP gametocytes at 8- and 15 min post-activation. At 8 min post-activation, many of the wildtype male gametocytes had progressed to the third genome division with multiple nuclear spindle poles within a central nucleus (*Figure 4Da*, *Figure 4—figure supplement 2A*). In many cases, a BB was visible in the cytoplasm closely associated with and connected to the NP (*Figure 4Db,c* and *Figure 4—figure supplement 2A,B*). From the NP, nuclear spindle microtubules (SMt) radiated into the nucleoplasm and connected to kinetochores (*Figure 4Db,c* and *Figure 4—figure supplement 2A,B*). Within the cytoplasm, several elongated axonemes, most but not all with the classical 9+2 microtubule arrangement, were visible (*Figure 4Dc*).

At 8 min in *Δsun1* gametocytes, there were predominantly mid-stage forms with elongated axonemes running around the nucleus (*Figure 4Dd* and *Figure 4—figure supplement 2C,D,G*). It was also possible to find clumps (groups) of BB (*Figure 4Dd,e* and *Figure 4—figure supplement 2E,F*). Electron densities similar to NPs were observed adjacent to the BB (*Figure 3De* and *Figure 4—figure supplement 2H,I,J*). An extensive search (100+sections) failed to identify examples of nuclear spindle formation with attached kinetochores. In contrast to the WT-GFP cells, a significant proportion of *Δsun1* cells (20%) had groups of centrally located kinetochores with no attached microtubules (naked kinetochores: NK) (*Figure 4Df* and *Figure 4—figure supplement 2C,D,G*). NK were not observed in

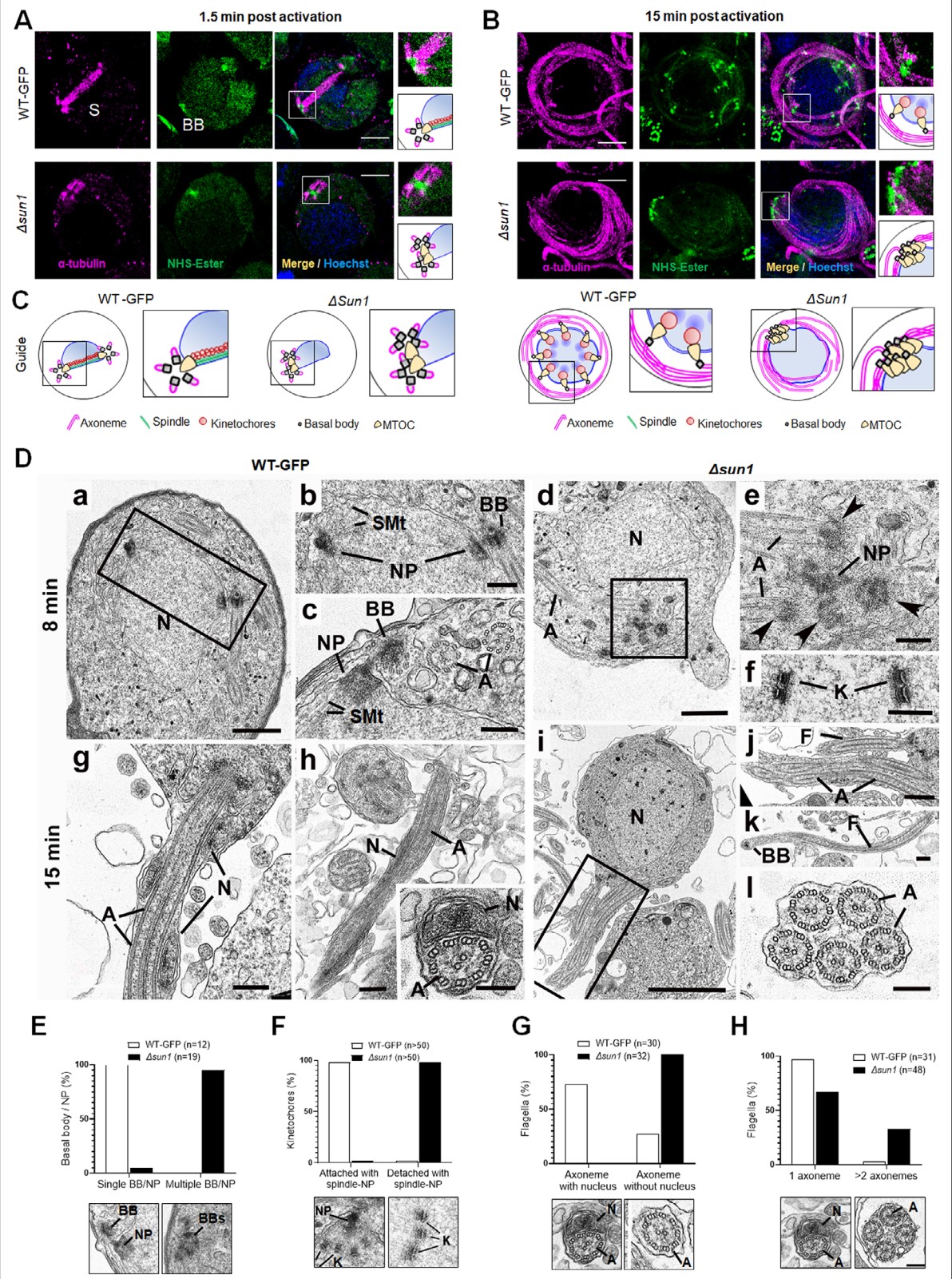

**Figure 4.** Ultrastructural analysis of *Δsun1* gametocytes showing defect in spindle formation and BB segregation. (**A**) Deletion of *sun*1 blocks first spindle formation as observed by expansion microscopy (ExM) of gametocytes activated for 1.5 min. α -tubulin: magenta, amine groups/NHS-ester reactive: green. Basal Bodies: BB; spindle: S. Insets represent the zoomed area marked by the white boxes shown around BB/microtubule-organising centre (MTOC) highlighted by NHS-ester and tubulin staining. Scale bar: 5 μm. (**B**) ExM images showing defect in BB/MTOC segregation in *Δsun1*

*Figure 4 continued on next page*

*Figure 4 continued*

gametocytes activated for 15 min. α -tubulin: magenta, amine groups/NHS-ester reactive: green. Basal Bodies: BB; Insets represent the zoomed area shown around BB/MTOC highlighted by NHS-ester and tubulin staining. More than 30 images were analysed in more than three different experiments. Scale bar: 5 µm. (**C**) The schematic illustrates structures associated with mitosis and axoneme formation showing the first spindle is not formed, and BB are not separated in *Δsun1* gametocytes. (**D**) Electron micrographs of WT-GFP microgametocytes (male) at 8 min (**a–c**) and 15 min (**g, h**) plus *Δsun1* gametocytes at 8 min (**d-f**) and 15 min (**i–l**). Bars represent 1 µm (**a, d, i**), and 100 nm (**b, c, e, f, g, h**, insert, **j, k, l**).(**a**) Low power magnification of WT-GFP microgametocyte showing the central nucleus (**N**) with two nuclear poles (**NP**) with a basal body (**BB**) adjacent to one. The cytoplasm contains several axonemes (**A**). (**b**) Enlargement of enclosed area (box) in (**a**) showing the BB adjacent to one NP. (**c**) Detail showing the close relationship between the NP and the BB. Note the cross-sectioned axonemes (**A**) showing the 9+2 microtubular arrangement. (**d**) Low power magnification of *Δsun1* cell showing a cluster of electron dense basal structures (enclosed area) in the cytoplasm adjacent to the nucleus (**N**). A – axonemes. (**e**) Detail from the cytoplasm (boxed area in **d**) shows a cluster of four basal bodies (arrowheads) and portions of axonemes (**A**) around a central electron dense structure of NP material. (**f**) Detail from a nucleus showing kinetochores (**K**) with no attached microtubules. (**g**) Periphery of a flagellating microgamete showing the flagellum and nucleus protruding from the microgametocyte. (**h**) Detail of a longitudinal section of a microgamete showing the spiral relationship between the axoneme (**A**) and nucleus (**N**). Insert. Cross-section of a microgamete showing 9+2 axoneme and adjacent nucleus (**N**). (**i**) Section through a microgametocyte with a central nucleus (**N**) undergoing exflagellation. (**j**) Enlargement of the enclosed area (box) in (**i**) showing one cytoplasmic protrusion containing a single axoneme forming a flagellum (**F**), while the other has multiple axonemes (**A**). (**k**) Longitudinal section through a flagellum (**F**) with a basal body (**B**) at the anterior end but note the absence of a nucleus. (**l**) Cross-section showing a cytoplasmic process contain five axonemes (**A**) but no associated nucleus. (**E–H**) Quantification of *Δsun1* phenotypes compared to WT-GFP during male gametogenesis. N=Nucleus; BB = Basal Body; NP = Nuclear pole; A=Axonemes.

The online version of this article includes the following figure supplement(s) for figure 4:

**Figure supplement 1.** Expansion Microscopy (ExM) shows defect in spindle formation and BB/microtubule-organising centre (MTOC) segregation during male gametogenesis in *Δsun1* parasites.

**Figure supplement 2.** Transmission Electron Microscopy (TEM) reveals defect in BB/microtubule-organising centre (MTOC) segregation and kinetochore attachment to spindle during male gametogenesis resulting in anucleate gamete formation in *Δsun1* parasites.

an extensive search of WT-GFP microgametocytes. (*Figure 4—figure supplement 2A*). Nevertheless, the parallel orientation of the axonemes appeared to be maintained and many displayed the normal 9+2 arrangement (*Figure 4Dc*).

By 15 min, WT-GFP gametocytes showed examples of exflagellation, with flagella displaying BB at the tip. Cross-sections showed gametes with normal axonemes with a 9+2 arrangement and an enclosed nucleus surrounded by a plasma membrane (*Figure 4Dg,h*). In *Δsun1* samples, several late stages as well as some mid-stages were observed. The late stages had more electron dense cytoplasm (*Figure 4Di*). The majority had few, if any axonemes, but all had a large nucleus with small clumps of electron dense material (*Figure 4Di*). A few microgametocytes appeared to be undergoing exflagellation with flagellar-like structures protruding from the cell surface, often with multiple axonemes (*Figure 4Dj,l*). However, they were abnormal, lacking a nucleus and often with variable numbers (1–6) of normal 9+2 axonemes (*Figure 4Dj,k,l*). There appeared to be a disconnect between the axonemes and the nucleus during exflagellation. There was evidence of chromatin condensation to form microgamete nuclei but no connection between the nucleus and axoneme during exflagellation, resulting in microgametes lacking a nucleus. This *Δsun1* mutant appeared to have a problem with separation of NPs and nuclear spindle formation, resulting in no genome segregation. The nuclear poles were difficult to find and did not appear to have been able to divide or move apart (*Figure 4Dd,e*, *Figure 4—figure supplement 2E,F,H,I,J*). It is possible that this lack of spindle pole division and movement was responsible for the BB remaining clumped together in the cytoplasm. It is also possible that clumping of the BB may explain the presence of multiple axonemes associated with the exflagellated structures. The quantification data for the *Δsun1* phenotypes are shown in *Figure 4E-H*, *Figure 4—figure supplement 2k*.

Together, these findings highlight the pivotal role of SUN1 in coordinating the nuclear and cytoplasmic compartments during male gametogenesis. Its absence disrupts spindle formation, BB segregation, and the physical attachment of axonemes to the nucleus, resulting in anucleate microgametes incapable of fertilisation.

## SUN1 interactome reveals associations with Nuclear envelope, ER, and chromatin components

To identify protein interaction partners of PbSUN1, we performed GFP-based immunoprecipitation (IP) using a nanobody targeting SUN1-GFP in lysates of purified gametocytes activated for 6–8 min.

This time point was chosen as it coincides with peak nuclear expansion and axoneme formation. To stabilise transient or weak interactions, protein cross-linking with 1% paraformaldehyde was used prior to IP. Co-immunoprecipitated proteins were identified using LC-MS/MS of tryptic peptides and were analysed using principal component analysis for duplicate WT-GFP and SUN1-GFP precipitations (*Figure 5A* and *Figure 5—source data 1*).

Co-variation with SUN1 was found for proteins of the nuclear envelope, chromatin, and ER-related membranes. Notably, SUN1 co-purified with nuclear pore proteins (NUP269, NUP335), membrane proteins, including a likely ER component (DDRGK-domain containing UFM1 E3 ligase, PBANKA_0927700), chromatin-related factors (e.g. condensin I/II subunits, topoisomerase II and the kinetochore subunit AKit-8), and the cytoplasmic and male gametocyte-specific kinesin-15 (PBANKA_145880). This suggests a role for SUN1 in bridging chromatin (through condensin II) on the nuclear side and the cytoskeleton (through kinesin-15) on the cytoplasmic side of the nuclear envelope, potentially in association with nuclear pores (*Figure 5A*) in a similar fashion to what has been proposed in the plant *Arabidopsis* (*Ito et al., 2024*; *Sakamoto et al., 2022*).

PbSUN1 also interacted with proteins harbouring a divergent carbohydrate-binding domain (like SUN1 itself), such as the allantoicase-like protein ALCC1 (*Sayers et al., 2024*), here referred to as ALLAN (PBANKA_1144200) and PBANKA_0209200, an ER-Golgi protein with a mannose-binding domain (*Figure 5A*, *Figure 5—source data 1*). ALLAN is an uncharacterised orthologue of allantoicase, an enzyme in purine metabolism. We could detect no KASH-like or lamin-like proteins in the co-immunoprecipitates. To further explore the interaction between ALLAN and SUN1, we performed the reciprocal ALLAN-GFP, from lysates of gametocytes 6 min post-activation. Amongst the interactors, we found SUN1 and its interactors, DDRGK-domain containing protein and kinesin-15 (*Figure 5B*, *Figure 5—source data 1*). The results from the immunoprecipitation experiments suggest that *Plasmodium* SUN1 functions in a non-canonical fashion via an interaction with ALLAN to tether chromatin to the nuclear envelope and possibly to the cytoplasmic cytoskeleton via kinesin-15 (*Figure 5C*).

We hypothesised that SUN1 is at the centre of multiple molecular interactions during male gametogenesis, coordinating NE remodelling, spindle organisation, and BB/axoneme attachment, with ALLAN as a key interactor near the nuclear MTOC.

## ALLAN is located at NP and influences basal body/NP segregation in male gametogenesis

To reveal ALLAN's function in the *Plasmodium* life cycle, we generated a C-terminal GFP-tagged ALLAN parasite line, confirming correct integration by PCR (*Figure 6—figure supplement 1A,B*) and detecting a protein of ~85 kDa by Western blot (*Figure 6—figure supplement 1C*). ALLAN-GFP parasites displayed normal growth and completed the life cycle, indicating that GFP tagging did not impair protein function.

During asexual blood stages, ALLAN-GFP exhibited a diffuse nucleoplasmic signal in trophozoites and schizonts with distinct focal points adjacent to dividing DNA, consistent with a role in mitotic regulation (*Figure 6—figure supplement 1D*). However, by late schizogony, the ALLAN signal had diminished. In male and female gametocytes (*Figure 6—figure supplement 1E*) and during zygote to ookinete transformation (*Figure 6—figure supplement 1F*), ALLAN-GFP showed a spherical distribution around Hoechst-stained nuclear DNA and was enriched to form distinct focal points. During oocyst development and liver schizogony, ALLAN-GFP also exhibited distinct focal points adjacent to dividing DNA (*Figure 6—figure supplement 1G,H*) suggesting a role in mitotic regulation, like in asexual blood stages (*Figure 6—figure supplement 1D*).

In activated male gametocytes, ALLAN-GFP rapidly localised within a minute post-activation to the NE, forming strong focal points that correlated with spindle poles or MTOCs (*Figure 6A* and *Video 2*). This localisation persisted as ploidy increased from 1N to 8N in successive rounds of genome replication (*Figure 6A*). Using U-ExM, ALLAN-GFP was resolved at the INM, enriched at spindle poles marked by NHS-ester staining, but absent from BB, where axonemes form (*Figure 6B and C*). The location of ALLAN-GFP showed no overlap with that of kinesin-8B, a BB marker, or that of kinetochore marker NDC80 (*Figure 6D and F*). Conversely, crosses with EB1-mCherry, a spindle marker, revealed an overlap at spindle poles (*Figure 6E*) suggesting a role for ALLAN in nuclear spindle pole organisation. Thus, the spatial relationship of SUN1 and ALLAN suggests coordinated roles in

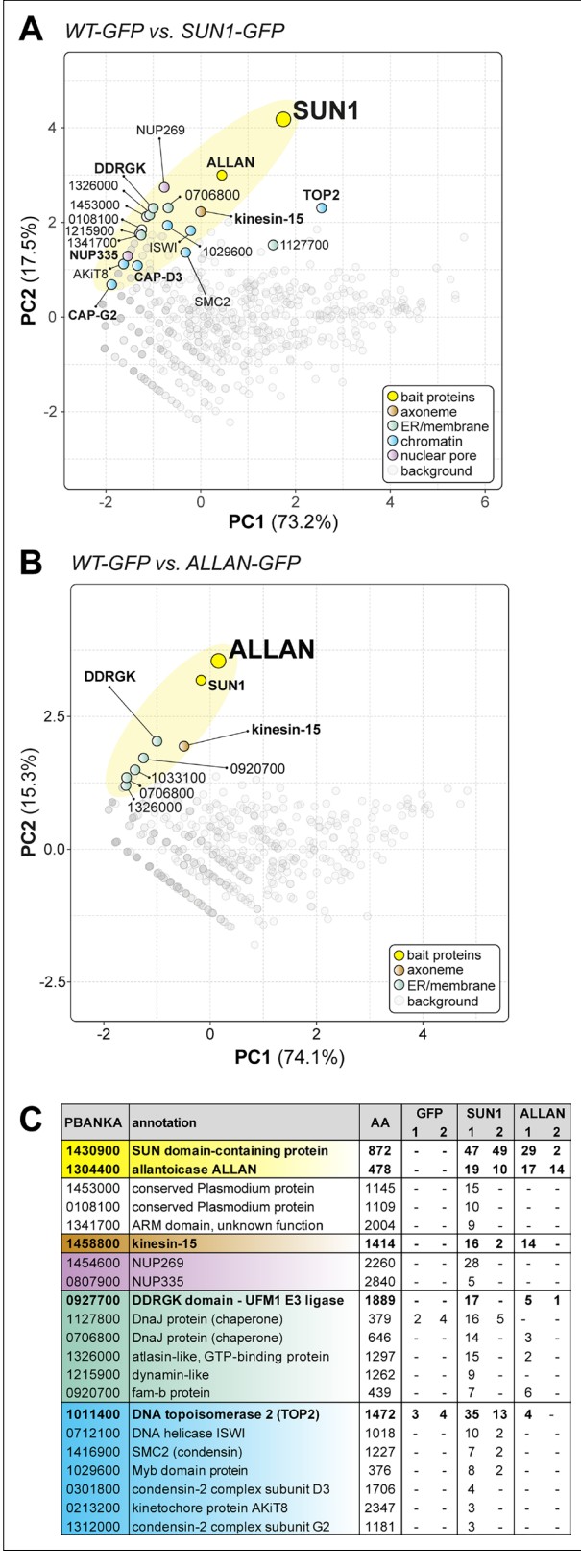

**C**

| PBANKA | annotation | AA | GFP 1 | GFP 2 | SUN1 1 | SUN1 2 | ALLAN 1 | ALLAN 2 |
|---|---|---|---|---|---|---|---|---|
| **1430900** | **SUN domain-containing protein** | **872** | - | - | **47** | **49** | **29** | **2** |
| **1304400** | **allantoicase ALLAN** | **478** | - | - | **19** | **10** | **17** | **14** |
| 1453000 | conserved Plasmodium protein | 1145 | - | - | 15 | - | - | - |
| 0108100 | conserved Plasmodium protein | 1109 | - | - | 10 | - | - | - |
| 1341700 | ARM domain, unknown function | 2004 | - | - | 9 | - | - | - |
| **1458800** | **kinesin-15** | **1414** | - | - | **16** | **2** | **14** | **-** |
| 1454600 | NUP269 | 2260 | - | - | 28 | - | - | - |
| 0807900 | NUP335 | 2840 | - | - | 5 | - | - | - |
| **0927700** | **DDRGK domain - UFM1 E3 ligase** | **1889** | - | - | **17** | **-** | **5** | **1** |
| 1127800 | DnaJ protein (chaperone) | 379 | 2 | 4 | 16 | 5 | - | - |
| 0706800 | DnaJ protein (chaperone) | 646 | - | - | 14 | 5 | 3 | - |
| 1326000 | atlasin-like, GTP-binding protein | 1297 | - | - | 15 | - | 2 | - |
| 1215900 | dynamin-like | 1262 | - | - | 9 | - | - | - |
| 0920700 | fam-b protein | 439 | - | - | 7 | - | 6 | - |
| **1011400** | **DNA topoisomerase 2 (TOP2)** | **1472** | **3** | **4** | **35** | **13** | **4** | **-** |
| 0712100 | DNA helicase ISWI | 1018 | - | - | 10 | 2 | - | - |
| 1416900 | SMC2 (condensin) | 1227 | - | - | 7 | 2 | - | - |
| 1029600 | Myb domain protein | 376 | - | - | 8 | 2 | - | - |
| 0301800 | condensin-2 complex subunit D3 | 1706 | - | - | 4 | - | - | - |
| 0213200 | kinetochore protein AKiT8 | 2347 | - | - | 3 | - | - | - |
| 1312000 | condensin-2 complex subunit G2 | 1181 | - | - | 3 | - | - | - |

**Figure 5.** Reciprocal co-immunoprecipitation of PbSUN1-GFP and ALLAN-GFP during male gametogony. (**A**) Projection of the first two components of a principal component analysis (PCA) of unique peptides derived from two SUN1-GFP (and WT-GFP) immunoprecipitations with GFP-trap (peptide values: *Figure 5—source data 1*). A subset of proteins is highlighted on the map based on relevant functional categories. (**B**) Similar to panel

*Figure 5 continued*

A, but now for the allantoicase-like protein ALLAN (PBANKA_1304400). (**C**) Selected proteins, their size and corresponding gene ID and representation by the number of peptides in either WT-GFP, PbSUN1-GFP, or ALLAN-GFP precipitates.

The online version of this article includes the following source data for figure 5:

**Source data 1.** List of proteins and numbers of unique peptides identified by proteomic analysis of GFP-trap-immunoprecipitates.

nuclear architecture and division of labour. SUN1 likely spans the NE, linking nuclear and cytoplasmic compartments, while ALLAN localises more specifically to nuclear MTOCs.

The proximity of both SUN1 and ALLAN at spindle poles is consistent with their collaboration to align spindle microtubules with nuclear and cytoplasmic MTOCs and ensure that chromosomal segregation aligns with axoneme formation. We performed functional studies of ALLAN deletion mutants to examine its contribution to spindle dynamics and its potential as a key player in rapid *Plasmodium* closed mitosis.

## Functional role of ALLAN in male gametogenesis

We generated *Δallan* mutants using double-crossover homologous recombination. Correct integration and successful gene deletion were confirmed via PCR (***Figure 7—figure supplement 1A,B***) and qRT-PCR, with no residual *allan* transcription in *Δallan* lines (***Figure 7—figure supplement 1C***). There was no phenotype in asexual blood stages, but *Δallan* mutants exhibited significant defects in mosquito stages.

*Δallan* parasites showed no marked reduction in male gamete exflagellation compared to WT-GFP controls (***Figure 7A***), and ookinete conversion rates were only slightly reduced (***Figure 7B***). However, oocyst counts were significantly lower in *Δallan*-parasite infected mosquitoes (***Figure 7C***), with diminished oocyst size (***Figure 7—figure supplement 1D***) and a significant decrease in sporozoite number at 14- and 21 d post-infection (***Figure 7D***). Though salivary gland sporozoites were significantly decreased (***Figure 7—figure supplement 1E***), *Δallan* parasites were successfully transmitted to mice in bite-back experiments, showing that some viable sporozoites were produced (***Figure 7E***). Analysis of DNA content (N) by fluorometric analyses after DAPI staining revealed that *Δallan1* male gametocytes were octaploid (8 N) at 8 min post-activation, similar to WT-GFP parasites (***Figure 7—figure supplement 1F***), indicating that the absence of ALLAN had no effect on DNA replication. We also checked for the presence of DNA in gametes stained with Hoechst (a DNA dye) and found that most *Δallan* gametes were anucleate (***Figure 7—figure supplement 1G***).

U-ExM analysis revealed a striking defect in *Δallan* male gametocyte morphology. At 8 min post-activation, NHS-ester staining indicated clustered BB, with incomplete segregation and misalignment relative to the nuclear MTOCs (***Figure 7F***). Despite normal axoneme elongation, spindle organisation was disrupted, as seen in non-segregated BB/MTOCs (***Figure 7F***).

To further explore the structural changes, we used electron microscopy analysis of gametocytes fixed at 8 min and 15 min post-activation (***Figure 7G***). At 8 min in the WT parasite, there was a large lobated nucleus exhibiting multiple NPs. Radiating from the NPs were microtubules forming the nuclear spindle with associated kinetochores. BB was closely associated with the NP on the cytoplasmic side from which extended an axoneme (***Figure 7Ga,b,c***). By 15 min, exflagellation had occurred and free microgametes were observed consisting of a 9+2 axoneme with closely associated nucleus enclosed by a unit membrane (plasmalemma) (***Figure 7Gd,e***).

In the *Δallan* parasites at 8 min, numerous mid- stage microgametocytes were observed. However, in contrast to the WT parasites, the BB appeared to be clumped together, although axoneme formation was similar to that in WT. (***Figure 7Gf,g***). NP-like structures lacking spindle microtubules were often associated with clumps of electron dense material (***Figure 7Gg***, ***Figure 7—figure supplement 2D***). Examination of sections of mid-stage gametocytes (50+), revealed several cells (~20%) with free kinetochores and no associated spindle microtubules (***Figure 6Gh*** and ***Figure 7—figure supplement 2A, B,C***). It was difficult to find individual NPs or obvious nuclear spindles, although very rare examples were observed (<2%).

By 15 min post-activation in *Δallan* gametocytes, several cells had exflagellated with the formation of numerous free gametes – the majority with no associated nucleus (***Figure 7Gi*** and ***Figure 7—figure***

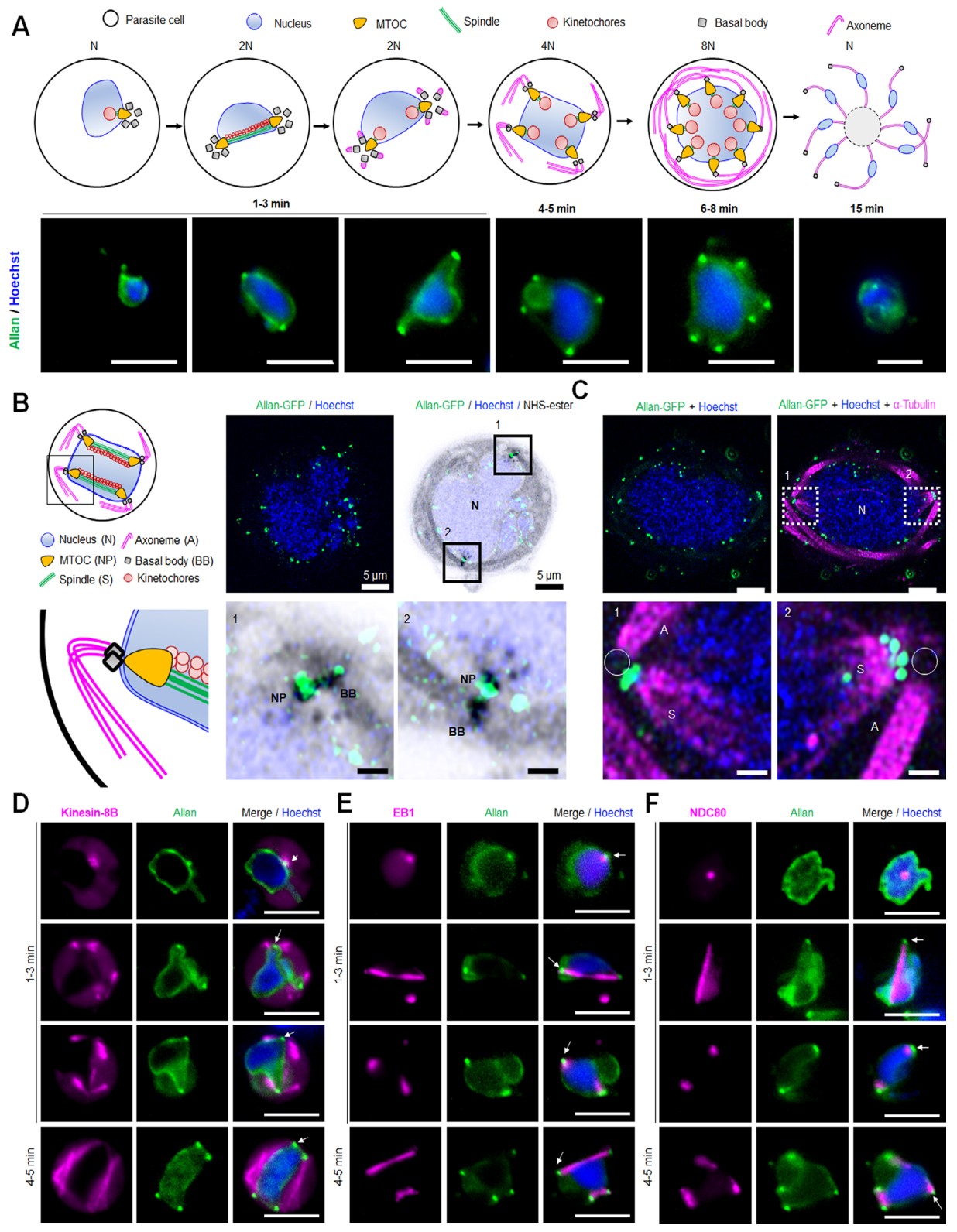

**Figure 6.** Location of ALLAN-GFP during male gametogenesis. (**A**) The schematic on the upper panel illustrates the process of male gametogenesis. N, ploidy of nucleus. Live cell images showing the location of ALLAN-GFP (green) at different time points (1–15 min) during male gametogenesis. Representative images of more than 50 cells with more than three biological replicates. Scale bar: 5 µm. (**B**) ExM images showing location of ALLAN-GFP (green) detected by anti-GFP antibody compared to nuclear pole (NP)/MTOC and BB stained with NHS ester (grey) in gametocytes activated for

*Figure 6 continued on next page*

*Figure 6 continued*

8 min. Scale bar: 5 µm. Representative images of more than 20 cells from two biological replicates. Insets represent the zoomed area shown around NP/MTOC and BB highlighted by NHS-ester. Scale bar: 1 µm. (**C**) ExM images showing the location of ALLAN-GFP (green) compared to spindle and axonemes (magenta) detected by anti-GFP and anti-tubulin staining, respectively, in gametocytes activated for 8 min. Representative images of more than 20 cells from two biological replicates. Scale: 5 µm. Insets represent the zoomed area shown around spindle/axonemes highlighted by tubulin and GFP staining. Basal Bodies: BB; Spindle: S; Axonemes: A; Nucleus: N. Scale bar: 1 µm. (**D, E, F**) Live cell imaging showing location of ALLAN-GFP (green) in relation to the BB and axoneme marker, kinesin-8B-mCherry (magenta) (**D**); spindle marker, EB1-mCherry (magenta) (**E**); and kinetochore marker, NDC80-mCherry (magenta) (**F**) during first mitotic division (1–3 min) of male gametogenesis. Arrows indicate the focal points of ALLAN-GFP. Representative images of more than 20 cells with three biological replicates. Scale bar: 5 µm.

The online version of this article includes the following source data and figure supplement(s) for figure 6:

**Figure supplement 1.** Generation of PbALLAN-GFP parasites and analysis of subcellular location of PbALLAN-GFP during various stages of parasite life cycle.

**Figure supplement 1—source data 1.** Tiff file of the original gel for *Figure 6—figure supplement 1B*, indicating the relevant band.

**Figure supplement 1—source data 2.** Tiff file of the original gel for *Figure 6—figure supplement 1B*.

**Figure supplement 1—source data 3.** Tiff file of the original gel for *Figure 6—figure supplement 1C*, indicating the relevant band.

**Figure supplement 1—source data 4.** Tiff file of the original gel for *Figure 6—figure supplement 1C*.

*supplement 2G,H*). Some had multiple axonemes (*Figure 7Gi,j* and *Figure 7—figure supplement 2H*) but there were many fewer examples than seen in the *Δsun1* mutant. However, unlike the *Δsun1* line, a few male gametes with axonemes and associated nuclei were observed. The *Δallan* phenotype quantitative data are shown in *Figure 7H–K* and *Figure 7—figure supplement 2I*.

These *Δallan* defects suggest that ALLAN is important for the separation of the NPs with formation of spindle microtubules and kinetochore attachment. The low incidence of spindle assembly and impaired NP organisation in *Δallan* gametocytes is reminiscent of the *Δsun1* mutant phenotype, reinforcing the relationship between these two proteins. The presence of a few normal nuclear spindle poles with attached kinetochores and formation of a few microgametes with axoneme and nucleus may explain the transmission observed for this mutant.

To further investigate the effect of *allan* deletion on transcription, we performed RNA-seq analysis in triplicate on *Δallan* gametocytes and WT-GFP gametocytes, at 8 min post-activation. A relatively small number of differentially expressed genes was identified (*Figure 7—figure supplement 1H*; *Figure 7—figure supplement 1—source data 3*). Gene ontology (GO)-based enrichment analysis of these genes showed that several upregulated genes coded proteins involved in either invasion, cell motility/gliding, or protein phosphorylation (*Figure 7—figure supplement 1I*).

## Evolution of a novel SUN1-ALLAN axis in haemosporida

The non-canonical SUN1-ALLAN complex in *P. berghei* is a unique apicomplexan adaptation, where NE and MTOC dynamics diverge from those of other eukaryotes. To trace the evolution of this interaction, we examined SUN1, ALLAN, and potential interactors (e.g. KASH proteins and lamins) across alveolates (apicomplexans, dinoflagellates and ciliates) and model organisms such as yeast, humans, and *Arabidopsis* (*Figure 8* and *Figure 8—source data 1*).

SUN domain proteins generally fall into two families: C-terminal SUN domain and N-terminal/mid SUN domain proteins (*Graumann et al., 2014*). Most eukaryotes have representatives of both families. *P. berghei* has two SUN domain proteins (*Kandelis-Shalev et al., 2024*), one of each family with PbSUN1 having a C-terminal SUN domain. Allantoicase-like proteins

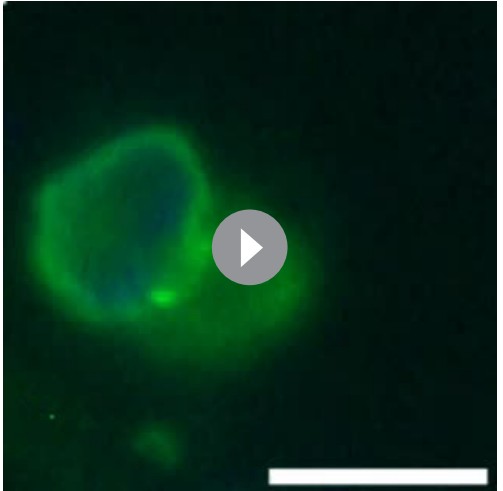

**Video 2.** Time-lapse video showing dynamic location ALLAN-GFP in gametocytes activated for 1-2 min. Scale: 5 µm.
https://elifesciences.org/articles/106537/figures#video2

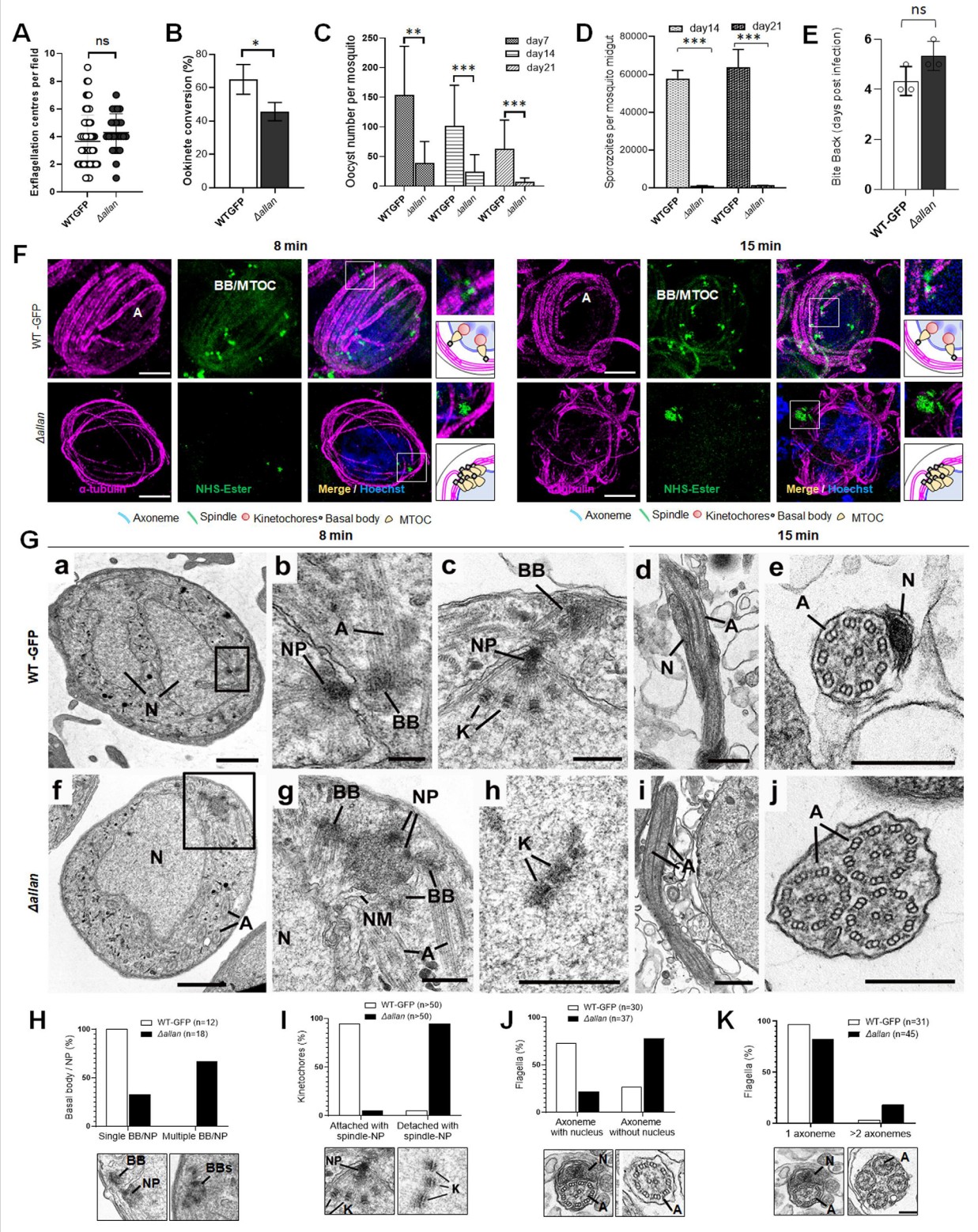

**Figure 7.** Deletion of ALLAN impairs male gametogenesis by blocking BB segregation. (**A**) Exflagellation centres per field at 15 min post-activation in *Δallan* compared to WT-GFP parasites. n≥3 independent experiments (>10 fields per experiment). Error bar ± SEM. (**B**) Percentage ookinete conversion from zygote. n≥3 independent experiments (>100 cells). Error bar ± SEM. (**C**) Total number of GFP-positive oocysts per infected mosquito in *Δallan* compared to WT-GFP parasites at 7-, 14-, and 21 d post-infection. Mean ± SEM. n≥3 independent experiments. (**D**) Total number of sporozoites in oocysts of *Δallan* compared to WT-GFP parasites at 14- and 21 d post-infection. Mean ± SEM. n≥3 independent experiments. (**E**) Bite back experiments

*Figure 7 continued on next page*

*Figure 7 continued*

reveal successful transmission of *Δallan* and WT-GFP parasites from mosquito to mouse. Mean ± SEM. n=3 independent experiments. Student's t-test and/or a two-way ANOVA test were employed to assess differences between control and experimental groups. Statistical significance is indicated as *P < 0.05, **P < 0.01, ***P < 0.001, or ns for not significant. (**F**) ExM images of gametocytes activated for 8- and 15 min showing MTOC/BB stained with NHS ester (green) and axonemes stained with anti-tubulin antibody (magenta). Axonemes: A; Basal Bodies: BB; Microtubule organising centre: MTOC. Insets represent the zoomed area shown around BB/MTOC highlighted by NHS-ester and tubulin staining. More than 30 images were analysed in more than three different experiments. Scale bar: 5 μm. (**G**) Electron micrographs of WT-GFP microgametocytes at 8 min (**a–c**) and 15 min (**d, e**) and the *Δallan* at 8 min (**f–h**) and 15 min (**I, j**). Bars represent 1 μm (**a, f**) and 200 nm in all other images. (**a**) Low power image of a microgametocyte showing the nucleus (**N**) with two NP complexes (arrows) consisting of the basal body, NP, and attached kinetochores. Axonemes (**A**) are present in the cytoplasm. (**b**) Enlargement showing the nuclear pole (NP), associated basal body (BB), and axonemes (**A**). (**c**) Detail of the nuclear pole (NP) showing kinetochores (**K**) attached to the spindle microtubules. Note the basal body (BB) adjacent to the nuclear pole (NP). (**d**) Longitudinal section of a microgamete showing the nucleus (**N**) closely associated with the axonemes (**A**). (**e**) Cross-section through a microgamete showing the nucleus (**N**) and axoneme (**A**) enclosed in plasma membrane. (**f**) Lower magnification of a microgametocyte showing a nucleus (**N**) with an adjacent clump of electron dense structures (enclosed area) and axonemes (**A**) in the cytoplasm. (**g**) Enlargement of the enclosed area in f showing multiple basal bodies (BB) and unseparated nuclear poles (NP) enclosed by portions of nuclear membrane (NM). N – nucleus. (**h**) Detail from a nucleus showing several kinetochores (**K**) with no associated spindle microtubules. (**i**) Longitudinal section of an exflagellating cytoplasmic process consisting of two axonemes (**A**) but no nucleus. (**j**) Cross-section through an exflagellating cytoplasmic process showing the presence of multiple axonemes (**A**) but the absence of any nucleus. (**H to K**) Quantification of *Δallan* phenotype compared to WT-GFP during male gametogenesis. N=Nucleus; BB = Basal Body; NP = Nuclear pole; A=Axonemes.

The online version of this article includes the following source data and figure supplement(s) for figure 7:

**Figure supplement 1.** Generation and genotype analysis of Pb*allan*-knockout (*Δallan*) parasites.

**Figure supplement 1—source data 1.** Tiff file of the original gel for *Figure 7—figure supplement 1B*, indicating the relevant band.

**Figure supplement 1—source data 2.** Tiff file of the original gel for *Figure 7—figure supplement 1B*.

**Figure supplement 1—source data 3.** List of genes differentially expressed between *Δallan vs* WT-GFP gametocytes activated for 8 min.

**Figure supplement 2.** Transmission electron microscopy (TEM) reveals defect in basal body/microtubule-organising centre (MTOC) segregation and kinetochore attachment to spindle during male gametogenesis resulting in anucleate gamete formation in *Δallan* parasites.

often consist of two nearly identical domains, belonging to the galactose-binding-like family that are remarkably similar to the SUN domain, (*Figure 8A*). Among apicomplexans, two allantoicase subtypes are present due to duplication in the ancestor of all Apicomplexa: AKiT8 (*Brusini et al., 2022*) a kinetochore-associated protein, and ALLAN, the allantoicase-like subtype. Notably, *Plasmodium* is one of the few apicomplexans retaining both subtypes (i.e. only AKiT is retained in Toxoplasma). Extreme sequence divergence complicates phylogenetic classification of allantoicase-like proteins among apicomplexans, but length-based classification distinguished the longer AKiT8 (~2300 aa) from ALLAN-like proteins (~800 aa; *Figure 8B*). Despite using previously developed Hidden Markov Model and structural searches (*Benz et al., 2024*), we found no evidence of KASH-domain proteins or lamins in amongst apicomplexans, similar to what was reported in prior studies (*Figure 8B*; *Koreny and Field, 2016*).

AlphaFold3 (AF3) modelling suggested that SUN1 and ALLAN interact specifically in Haemosporida through the N-terminal domain of SUN1, a region that is absent from other apicomplexan and/or other eukaryotic SUN1 family proteins (*Figure 8C*). We could not detect similar interactions between SUN1-like and ALLAN-like orthologs amongst other apicomplexans, ruling out a rapidly evolving interaction. Based on these structural predictions and the super-resolution imaging of SUN1- and ALLAN-GFP lines (*Figures 2D, E ,, 6B and C*), we propose that ALLAN resides on the nucleoplasmic side of the NE, while SUN1's C-terminal domain likely extends toward the ONM to interact with proteins involved in axoneme and cytoskeletal dynamics in the nucleoplasm (*Figure 8D*). This specificity may underscore the restricted nature of a SUN1-ALLAN complex to Haemosporida, and thus specific adaptation of the allantoicase-like protein ALLAN for nuclear envelope dynamics.

## Discussion

This study identifies the SUN1-ALLAN complex as a novel and essential mediator of NE remodelling and bipartite MTOC coordination during *P. berghei* male gametogenesis. *Plasmodium* lacks KASH-domain proteins and lamins, and a canonical LINC complex, relying on a highly divergent mechanism to tether nuclear and cytoplasmic compartments during rapid closed mitosis (*Rout et al., 2017*).

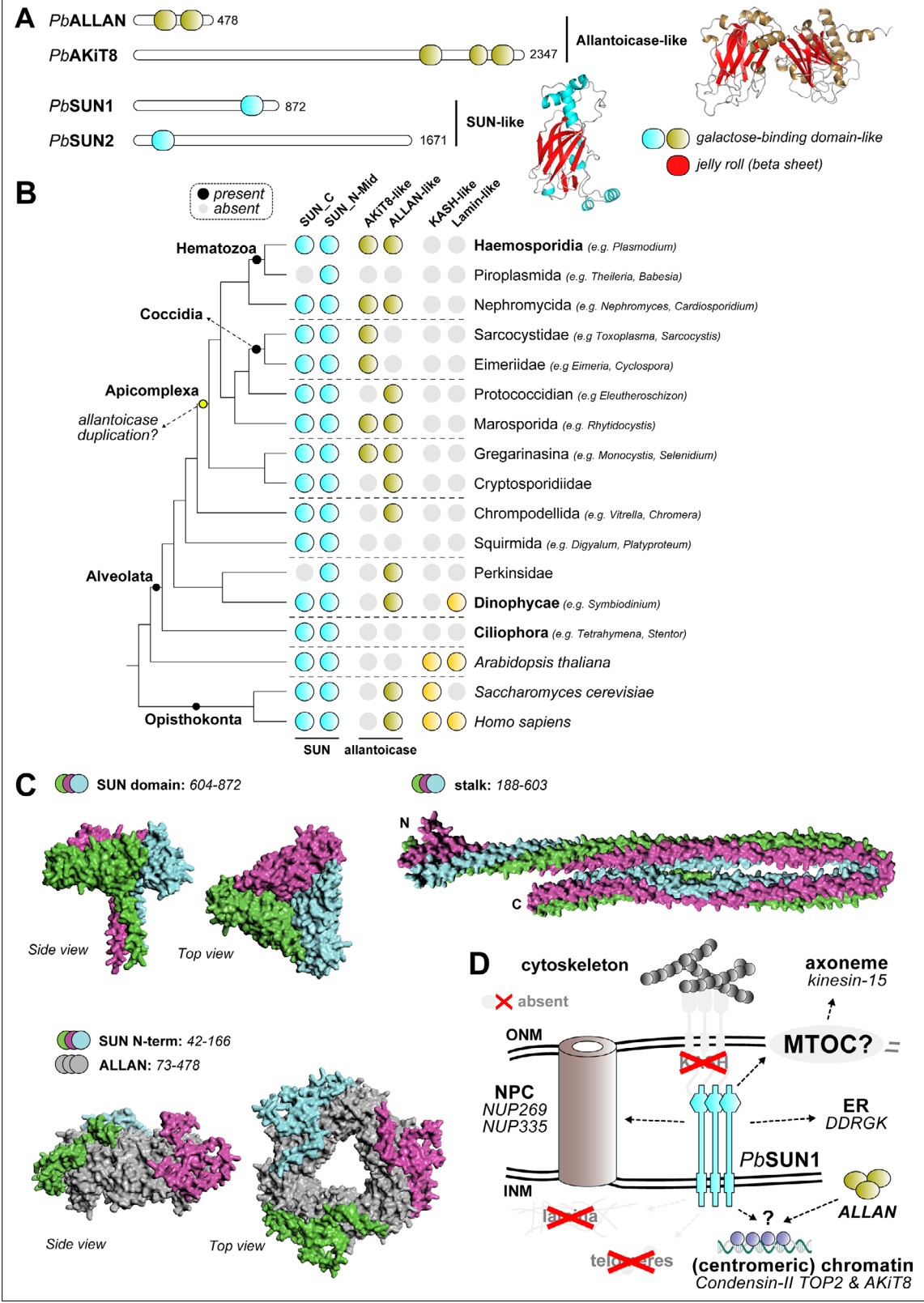

**Figure 8.** Evolution and Structure of the SUN1-ALLAN interaction. (**A**) Domain analysis shows two proteins with allantoicase domain and two proteins with SUN-domain in *P. berghei*. The SUN domain and two domains comprising allantoicases are part of the same galactose-binding domain family, with a strikingly similar fold. (**B**) Phylogenetic profiles showing the presence of SUN-, ALLAN- and KASH-domain and lamin proteins in Apicomplexa and a selection of other eukaryotes, including two model species *Homo sapiens* and *Arabidopsis thaliana*. (**C**) AlphaFold3-modelled interaction between

*Figure 8 continued on next page*

*Figure 8 continued*

ALLAN and SUN1 based on separate domains (no full structure could be modelled). The SUN1 C-terminus forms a trimeric complex (pTM:0.37) similar to a trimeric ALLAN complex (grey) with the N-terminus of SUN1 interacting with ALLAN (pTM:0.55). This N-terminal domain is unique to Haemosporida. (D) Overview in similar style as *Figure 1* of main interactors for putative localisation at the nuclear envelope for ALLAN and SUN1 during male gametogenesis. Structures in grey have not been found to be associated with SUN1.

The online version of this article includes the following source data for figure 8:

**Source data 1.** Phylogenetic analysis of SUN1, allantoicase-like, KASH, and lamin proteins in Eukaryotes with a focus on Apicomplexa with sources for genomes (hyperlinks), sequences, and IDs.

Our results reveal that SUN1 and ALLAN form a unique complex essential for spindle assembly, BB segregation, and axoneme organisation, maintaining an intact NE throughout. Disruption of either protein causes spindle and MTOC miscoordination, leading to defective male gametes incapable of fertilisation. Additionally, SUN1 deletion alters lipid homeostasis and NE dynamics, underscoring its multifaceted role in nuclear organisation and metabolism. These findings establish the SUN1-ALLAN axis as a crucial evolutionary innovation in *Plasmodium*.

In *P. berghei* male gametogenesis, mitosis is closed and extraordinarily rapid with features that diverge significantly from those of classical models. Within just 8 min, there are three rounds of DNA replication (1N to 8N), accompanied by the assembly of axonemes on cytoplasmic BB (*Guttery et al., 2022*). KASH-domain proteins and lamins are absent, yet *Plasmodium* achieves precise nuclear and cytoskeletal coordination. Our findings reveal that this is accomplished through a non-canonical SUN1-ALLAN complex. SUN1 located at the NE forms loops and folds to accommodate rapid nuclear expansion, and ALLAN facilitates spindle pole coordination.

Recent studies on SUN1-domain containing proteins have been reported in some Apicomplexa parasites including *Toxoplasma* and *Plasmodium* (*Kandelis-Shalev et al., 2024*; *Sayers et al., 2024*; *Wagner et al., 2023*). SUN-like protein-1 of *Toxoplasma gondii* shows a mitotic spindle pole (MTOC) location and is important for nuclear division (*Wagner et al., 2023*). *Plasmodium spp.* encode two SUN proteins (SUN1 and SUN2), that show a nuclear membrane location and play roles in nuclear division and DNA repair during the blood stage of *P. falciparum* (*Kandelis-Shalev et al., 2024*). Recent work by *Sayers et al., 2024* demonstrated that PbSUN1-HA was associated with the NE and suggested that it may be required to capture the intranuclear MTOC onto the NE (*Sayers et al., 2024*). They also showed that it forms a complex with ALCC1 (the protein we have designated ALLAN). They showed that SUN1 is essential for fertile male gamete formation and that in SUN1-KO lines basal bodies are aggregated. Our study extends their findings by providing live-cell imaging of SUN1 and ALLAN and their coordination with key mitotic markers such as NDC80, EB1, and Kinesin-8B. We generated and characterised both SUN1 and ALLAN knockout parasites, performing ultrastructural analyses using TEM and extensive functional studies. Consistent with the results of Sayers et al., we observed basal body clumping in SUN1-KO, and showed that kinetochore attachment to the spindle is compromised in both mutant parasites. Our work emphasises the role of the nuclear envelope in the rapid cell division during male gametogenesis. Collectively, the findings from both studies highlight the importance of the SUN1–ALLAN complex in Plasmodium cell division (*Sayers et al., 2024*).

*Plasmodium* SUN1 shares some similarities with SUN proteins in other organisms like *S. cerevisiae and Schizosaccharomyces pombe* (*Fan et al., 2022*; *Hagan and Yanagida, 1995*), but its interaction with ALLAN is a highly specialised adaptation in *Plasmodium*. The location of SUN1 at the nuclear MTOC, distinct from axonemal markers like kinesin-8B, underscores its role in nuclear compartmentalisation. Similarly, the absence of functional spindle formation and kinetochore attachment in *Δsun1* mutants is reminiscent of phenotypes seen in mutants of spindle-associated proteins in *Plasmodium*, such as EB1, ARK2, and kinesin-8X (*Zeeshan et al., 2023*; *Zeeshan et al., 2019b*). However, SUN1's role extends specifically to NE remodelling and the organisation of the inner acentriolar MTOC.

Our study demonstrates that SUN1 is primarily located in the space between the INM and ONM, with the NE but facing the nucleoplasm via it's N-terminus, forming dynamic loops and folds to accommodate the rapid expansion of the nucleus during three rounds of genome replication. These loops serve as structural hubs for spindle assembly and kinetochore attachment at the nuclear MTOC, separating nuclear and cytoplasmic compartments. The absence of SUN1 disrupts spindle formation, causing defects in chromosome segregation and kinetochore attachment, while BB segregation

remains incomplete, resulting in clumped basal bodies and anuclear flagellated gametes. This phenotype highlights the role of SUN1 in maintaining the nuclear compartment of the bipartite MTOC.

ALLAN, a novel allantoicase-like protein, has a location complementary to that of SUN1, forming focal points at the inner side of the NE. Like SUN1, ALLAN is also not essential during asexual erythrocytic stage proliferation. Functional studies following ALLAN gene deletion revealed similar defects in MTOC organisation, with impaired spindle formation, kinetochore attachment, and nuclear-cytoplasmic coordination during flagellated gamete assembly. Compared to SUN1, ALLAN gene deletion exhibited a less stringent phenotype because ALLAN knock-out parasites were able to transmit the disease from vector to host, while SUN1 knock-out parasites were blocked. This suggests that SUN1 has additional functional roles. These findings indicate that SUN1 and ALLAN work together, replacing the role of lamins and KASH proteins in coordinating nuclear and cytoskeletal dynamics.

Interestingly, our interactome analysis identified kinesin-15 as a putative interactor of SUN1 and ALLAN, suggesting a possible link between this motor protein and nuclear remodelling. Previous studies have shown that kinesin-15 is essential for male gamete formation and is located partially at the plasma membrane and nuclear periphery, albeit mostly cytoplasmic (*Zeeshan et al., 2022b*). On the nuclear side, we found subunits of the condensin-II complex, similar to what was found in *Arabidopsis* (*Ito et al., 2024*; *Sakamoto et al., 2022*). These interactions with SUN1 and ALLAN raise the possibility of a broader network of proteins facilitating nuclear and cytoskeletal coordination during male gametogenesis.

Transcriptomic analysis of the *Δsun1* mutant showed upregulation of genes involved in lipid metabolism and microtubule organisation. Lipidomic profiling further confirmed substantial alterations in the lipid composition of SUN1-deficient gametocytes. Notably, PA and CE, both critical for membrane curvature and expansion, were reduced in the *Δsun1* mutant, whereas MAG and DAG levels were elevated. These disruptions likely reflect impaired NE expansion and structural integrity during gametogenesis. Additionally, the altered levels of specific fatty acids, such as arachidonic acid and myristic acid, suggest perturbations in both scavenged and apicoplast-derived lipid pathways.

The SUN1-ALLAN complex exemplifies a lineage-specific adaptation in *P. berghei*, highlighting the evolutionary plasticity of NE remodelling mechanisms across eukaryotes. SUN-domain proteins, a hallmark of LINC complexes, are widely conserved and diversified across eukaryotic lineages, yet *Plasmodium* and related apicomplexans lack key canonical LINC components.

Our analyses suggest that the SUN1-ALLAN complex is specific to Haemosporida. While SUN-domain proteins are conserved across most eukaryotes, the allantoicase-like protein ALLAN is a novel addition to the nuclear architecture of apicomplexans. Phylogenetic profiling indicates that ALLAN likely arose from a duplication of an allantoicase in an early apicomplexan ancestor, likely the ancestor to all coccidians and hematozoa. This event gave rise to two subtypes: AKiT8, associated with kinetochores (*Brusini et al., 2022*), and ALLAN, specialised in NE dynamics.

The absence of KASH proteins and lamins in apicomplexans raises fascinating questions about how NE and cytoskeletal coordination evolved in this group. Comparative studies in ichthyosporeans, close relatives of fungi and animals, have identified intermediate forms of mitosis involving partial NE disassembly (*Shah et al., 2024*). In contrast, *Plasmodium* employs a highly streamlined closed mitosis, likely driven by the requirement for rapid nuclear and cytoskeletal coordination during male gametogenesis. The discovery of the SUN1-ALLAN axis suggests that apicomplexans have developed alternative strategies to adapt the NE for their cellular and mitotic requirements. These findings raise intriguing questions about the evolutionary pressures that shaped this unique nuclear-cytoskeletal interface in apicomplexans. The presence of similar or other non-canonical complexes in other organisms remains an open question, with potential implications for understanding the diversity of mitotic processes.

This study opens several avenues for future research into the biology of the malaria parasite and its unique adaptations for mitosis. While the SUN1-ALLAN complex has been shown to play a central

role in bipartite MTOC organisation during male gametogenesis, many questions remain about its broader role and potential applications. Another key question concerns the evolutionary origins of the SUN1-ALLAN axis; comparative studies across apicomplexans, dinoflagellates, ciliates, and other non-model eukaryotes may help trace the evolution of this complex. The presence of similar non-canonical complexes in other rapidly dividing eukaryotic systems may shed light on the diversity of mitotic adaptations beyond *Plasmodium*. A functional further exploration of associated proteins, such as kinesin-15, may help elucidate how the SUN1-ALLAN complex integrates with other cellular machinery and structures. Investigating whether kinesin-15, a motor protein, interacts directly with SUN1 or ALLAN and has a catalytic role in nuclear-cytoskeletal remodelling may be critical to further our understanding of how the system achieves such rapid mitotic coordination.

# Methods

## Key resources table

| Reagent type (species) or resource | Designation | Source or reference | Identifiers | Additional information |
|---|---|---|---|---|
| Strain, strain background (*Escherichia coli*) | X2-Blue ultracompetent cells | Agilent Technologies | Cat #200150 | chemically competent cells |
| Cell line (*P. berghei*) | ANKA 2.34 | | WT-ANKA RRID:NCBITaxon_5823 | |
| Cell line (*P. berghei*) | 507cl1 | | WT-GFP | |
| Cell line (*P. berghei*) | SUN1-GFP | This study | | Tag line |
| Cell line (*P. berghei*) | ALLAN-GFP | This study | | Tag line |
| Cell line (*P. berghei*) | kinesin-8B-mCherry | | | Tag line |
| Cell line (*P. berghei*) | EB1-mCherry | | | Tag line |
| Cell line (*P. berghei*) | NDC80-mCherry | | | Tag line |
| Cell line (*P. berghei*) | ARK2-mCherry | | | Tag line |
| Cell line (*P. berghei*) | Δsun1 | This study | | KO line |
| Cell line (*P. berghei*) | Δallan | This study | | KO line |
| Cell line (*P. berghei*) | Δnek4 | | | KO line |
| Cell line (*P. berghei*) | Δhap2 | | | KO line |
| Antibody | Monoclonal Anti-α-Tubulin antibody from mouse (DM1A) | Sigma | Cat# T9026 RRID:AB_477593 | IF, UExM (1:1000) |
| Antibody | Rabbit anti-GFP Polyclonal Antibody | Thermo Fisher | Cat# A-11122 RRID:AB_221569 | WB, UExM (1:1250) |
| Antibody | Goat anti-Rabbit IgG Alexa Fluor 568 (polyclonal) | Thermo Fisher | Cat# A11036 RRID:AB_10563566 | IF, UExM (1:1000) |
| Antibody | Goat anti-Mouse IgG Alexa Fluor 568 (polyclonal) | Thermo Fisher | Cat# A-11004 RRID:AB_2534072 | IF, UExM (1:1000) |
| Antibody | Goat anti-Rabbit IgG Alexa Fluor 488 (polyclonal) | Thermo Fisher | Cat# A-11034 RRID:AB_2576217 | IF, UExM (1:1000) |
| Antibody | Cy3-conjugated mouse monoclonal antibody 13.1 | N/A | | Live-cell imaging (1:1000) |
| Chemical compound | Atto 594 NHS ester | Merck | Cat #08741 | UExM (10 µg/ml) |

## Generation of transgenic parasites and genotype analyses

To generate lines for GFP-tagged SUN1 and ALLAN, a region of each gene downstream of the ATG start codon was amplified, ligated to the p277 vector, and transfected as previously described (*Guttery et al., 2014*). The p277 vector includes a human DHFR cassette, providing resistance to pyrimethamine. Schematic representations of the endogenous gene loci, the vector constructs, and the recombined gene loci can be found in *Figure 2—figure supplement 1A* and *Figure 6—figure supplement 1A*. For parasites expressing C-terminal GFP-tagged proteins, diagnostic PCR was performed with primer 1 (Int primer) and primer 2 (ol492) to confirm integration of the GFP -targeting

**Table 1.** Oligonucleotides used in this study.

| Name | Sequence (5' to 3') | Notes |
|---|---|---|
| **Primers used for GFP tag construct** | | |
| T3031 (SUN1) | CCCCGGTACCGAAAGTGGTAATGTATCTGAAAC | KpnI site underlined |
| T3032 (SUN1) | CCCCGGGCCCCTTTAACTTTCTTATGCATCTTTGAC | ApaI site underlined |
| Int303 (SUN1) | CAAGAATTGTTCGATGGCATG | |
| T3071 (Allan) | CCCCGGTACCGAGGTTGTAGAAAATCCCTG | KpnI site underlined |
| T3072 (Allan) | CCCCGGGCCCAGGGGGGGATTGATAAAAC | ApaI site underlined |
| Int307 (Allan) | GCATATGCCTATAGTAATTCGTG | |
| ol492 | ACGCTGAACTTGTGGCCG | |
| mCherryP | TTCAGCTTGGCGGTCTGGGT | |
| **Primers used for knockout construct** | | |
| N1511 (SUN1) | CCCCGGGCCCAGGGGAAAGCAACAGCATTG | ApaI site underlined |
| N1512 (SUN1) | GGGGAAGCTTCCCTTTCTGCCCTTTGGTTT | HindIII site underlined |
| N1513 (SUN1) | CCCCGAATTCACGGAAAACTATGGTGCCCC | EcoRI site underlined |
| N1514 (SUN1) | GGGGTCTAGAACGCCCTTTTACTCCTATCTACA | XbaI site underlined |
| intN151_5 (SUN1) | GCACATTTGATCGTATACATGAC | |
| N1531 (Allan) | CCCCGGGCCCGTTCGTATCTCCATAATTATTAAAGG | ApaI site underlined |
| N1532 (Allan) | GGGGAAGCTTCGTTAATATTTTTCTTCGCCG | HindIII site underlined |
| N1533 (Allan) | CCCCGAATTCGATTGGTTTCAATTACCTCCTTG | EcoRI site underlined |
| N1534 (Allan) | GGGGTCTAGACTATATATGCGCAGGGATATAC | XbaI site underlined |
| intN153_5 (Allan) | GTGTAATAGCCATCATAATTAAGC | |
| ol248 | GATGTGTTATGTGATTAATTCATACAC | |
| **Primers used for qRT PCR** | | |
| hsp70 FW | GTATTATTAATGAACCCACCGCT | |
| hsp70 RV | GAAACATCAAATGTACCACCTCC | PBANKA_081890 |
| arginyl-tRNA FW | TTGATTCATGTTGGATTTGGCT | |
| arginyl-tRNA RV | ATCCTTCTTTGCCCTTTCAG | PBANKA_143420 |
| Allantoicase-FW | ACCCTTGATTCCTCATGTCTTCAA | |
| Allantoicase-RV | TTTTTCCTGAGCCGGTTGCT | PBANKA_1304400 |
| SUN1-FW | GGGCTCTAGAATCATTAGGAGC | |
| SUN1-RV | TCTCCTGGGAAGTTTGAAGGT | PBANKA_1430900 |

constructs (*Figure 2—figure supplement 1B* and *Figure 6—figure supplement 1B*). The primers used to generate the mutant parasite lines can be found in *Table 1*.

Gene-deletion targeting vectors for SUN1, and ALLAN were created using the pBS-DHFR plasmid. This plasmid contains polylinker sites flanking a *Toxoplasma gondii* dhfr/ts expression cassette, which provides resistance to pyrimethamine, as described previously (*Saini et al., 2017*). PCR primers N1511 and N1512 were used to amplify a 1,1094 bp fragment of the 5′ sequence upstream of *sun1* from genomic DNA, which was then inserted into the ApaI and HindIII restriction sites upstream of the dhfr/ts cassette in the pBS-DHFR plasmid. A 776 bp fragment from the 3′ flanking region of *sun1* was generated with primers N1513 and N1514, and inserted downstream of the dhfr/ts cassette using the

EcoRI and XbaI restriction sites. The same approach was used for *Allan*, amplifying upstream (1044 bp) and downstream (1034 bp) sequences and insertion into the pBS-DHFR plasmid. The linear targeting sequence was released from the plasmid using *ApaI/XbaI* digestion. A schematic representation of the endogenous *sun1* and *allan* loci, the construct and the recombined *sun1* and *allan* loci are presented in *Figure 3—figure supplement 1A* and *Figure 7—figure supplement 1A*, respectively. The primers used to generate these mutant parasite lines can be found in *Table 1*. A diagnostic PCR used primer 1 (IntN151_5) and primer 2 (ol248) to confirm integration of the targeting construct, and primer 3 (Int303) and primer 4 (N1514) were used to confirm deletion of the *sun1* gene (*Figure 2—figure supplement 2B* and *Table 1*). Similarly, a diagnostic PCR using primer 1 (IntN153_5) and primer 2 (ol248) was used to confirm integration of the targeting construct, and primer 3 (Int307) and primer 4 (N1534) were used to confirm deletion of the *allan* gene (*Figure 7—figure supplement 1B* and *Table 1*). *P. berghei* ANKA line 2.34 (for GFP-tagging), and ANKA line 507cl1 expressing GFP (for the gene-deletion) were transfected by electroporation (*Janse et al., 2006*).

## Purification of schizonts and gametocytes

Blood cells obtained from infected mice (day 4 post-infection) were cultured for 11 hr and 24 hr at 37°C (with rotation at 100 rpm) and schizonts were purified the following day on a 60% v/v NycoDenz (in phosphate-buffered saline (PBS)) gradient, (NycoDenz stock solution: 27.6% w/v NycoDenz in 5 mM Tris-HCl (pH 7.20), 3 mM KCl, 0.3 mM EDTA).

Gametocytes were purified using a modified version of the protocol of *Beetsma et al., 1998*. In brief, parasites were injected into mice pre-treated with phenylhydrazine and enriched by sulfadiazine treatment 2 d post-infection. Blood was collected on the fourth day post-infection, and gametocyte-infected cells were purified using a 48% NycoDenz gradient (prepared with NycoDenz stock solution as above). Gametocytes were harvested from the gradient interface and washed thoroughly.

## Live-cell imaging

To investigate SUN1-GFP, and ALLAN-GFP expression during erythrocytic stages, parasites cultured in schizont medium were imaged at various stages of schizogony. Purified gametocytes were assessed for GFP expression and location at different time points (0 and 1 to 15 min) post-activation in ookinete medium (RPMI 1640 medium containing 100 μM xanthurenic acid, 1% w/v sodium bicarbonate, and 20 % v/v heat inactivated foetal bovine serum [FBS]). Zygote and ookinete stages were labelled using the cy3-conjugated mouse monoclonal antibody 13.1 (red), which targets the P28 surface protein (*Winger et al., 1988*), and examined over a 24-hr period. Oocysts and sporozoites were imaged in infected mosquito guts. All images were captured with a 63x oil immersion objective on a Zeiss Axio-Imager M2 microscope equipped with an AxioCam ICc1 digital camera.

## Generation of dual-tagged parasite lines and co-localisation analysis

The GFP lines of SUN1 or ALLAN parasites were mixed with mCherry lines of either NDC80 or EB1 or kinesin-8B or ARK2 in equal numbers and injected into a mouse. Mosquitoes were fed on this mouse 4–5 d after infection when gametocyte parasitemia was high. These mosquitoes were checked for oocyst development and sporozoite formation at day 14 and day 21 after feeding. Infected mosquitoes were then allowed to feed on naïve mice, and after 4–5 d, these mice were examined for blood stage parasitaemia by microscopy with Giemsa-stained blood smears. In this way, some parasites expressed both GFP and mCherry in the resultant gametocytes, and these were purified, and fluorescence microscopy images were collected as described above.

Co-localisation between proteins was assessed in ImageJ. A line was drawn over a two-channel image and the line added to the ROI manager in the same place on both channels. The multiplot function was then used to create a plot of the intensity along the line in both channels of the image. The values from the plot were copied into Excel and the Pearson's correlation coefficient calculated as a representation of co-localisation.

## Western blot analysis

Purified gametocytes were placed in lysis buffer (10 mM Tris-HCl [pH 7.5], 150 mM NaCl, 0.5 mM EDTA, and 1% NP-40). The lysed samples were placed for 10 min at 95°C after adding Laemmli buffer, and then centrifuged at 13,000 g for 5 min. Samples were electrophoresed on a 4% to 12%

SDS-polyacrylamide gel, and resolved proteins were transferred to nitrocellulose membrane (Amersham Biosciences). Immunoblotting was performed using the Western Breeze Chemiluminescence Anti-Rabbit kit (Invitrogen) and an anti-GFP polyclonal antibody (Invitrogen) at a dilution of 1:1250, according to the manufacturer's instructions.

## Generation of dual-tagged parasite lines

The GFP (green)-tagged SUN1, or ALLAN parasites were mixed in equal numbers with mCherry (red)-tagged lines of kinetochore marker (NDC80), basal body/axoneme marker (kinesin-8B), and spindle markers (EB1 and ARK2) and injected into mice. Mosquitoes fed on these mice 4 to 5 d post-infection, when gametocytaemia was high, were monitored for oocyst development and sporozoite formation at days 14 and 21 after feeding. Infected mosquitoes were then allowed to feed on naïve mice, and after 4 to 5 d, these mice were examined for blood-stage parasitaemia by microscopy of Giemsa-stained blood smears. Some parasites expressed both GFP- and mCherry-tagged proteins in the resultant gametocytes; these cells were purified, and fluorescence microscopy images were collected as described above.

## Parasite phenotype analyses

Blood samples containing approximately 50,000 SUN1-knockout, or ALLAN-knockout parasites were injected intraperitoneally (i.p.) into mice. Asexual stage parasite development and gametocyte production were monitored by microscopy on Giemsa-stained thin smears. Four to five days post-infection, gametocytes were harvested and exflagellation and ookinete conversion were examined as described above, using a Zeiss AxioImager M2 microscope. To analyse mosquito infection and parasite transmission, 30 to 50 *Anopheles stephensi* SD 500 mosquitoes were allowed to feed for 20 min on anaesthetised, infected mice that had at least 15% asexual parasitaemia and a comparable gametocyte level. To assess midgut infection, approximately 15 guts were dissected from mosquitoes on days 7 and 14 post-feeding, and oocysts were counted by microscopy using a 63× oil immersion objective. On day 21 post-feeding, another 20 mosquitoes were dissected, and their guts and salivary glands were crushed separately in a loosely fitting homogeniser to release sporozoites, which were then quantified using a haemocytometer or used for imaging.

Mosquito bite-back experiments with naïve mice were conducted 21 d post-feeding, and blood smears were examined after 3 to 4 d.

## Immunoprecipitation and mass spectrometry

Purified male gametocyte pellets from SUN1-GFP, and ALLAN-GFP parasites at 6-min post-activation were cross-linked by 10 min incubation with 1% formaldehyde, followed by 5 min incubation in 0.125 M glycine solution and three washes with phosphate-buffered saline [PBS; pH 7.5]. WT-GFP gametocytes were used as controls. Cell lysates were prepared and immunoprecipitation was conducted using a GFP-Trap_A Kit (Chromotek) according to the manufacturer's instructions. Briefly, lysates were incubated for 2 hr with GFP-Trap_A beads at 4°C with continuous rotation, then unbound proteins were washed away, and bound proteins were digested with trypsin. The tryptic peptides were analysed by liquid chromatography-tandem mass spectrometry. Mascot (http://www.matrixscience.com/) and MaxQuant (https://www.maxquant.org/) search engines were used for mass spectrometry data analysis. Experiments were performed in duplicate. Peptides and proteins with a minimum threshold of 95% were used for further proteomic analysis. The PlasmoDB database was used for protein annotation, and manual curation was used to classify proteins into categories relevant for SUN1/ALLAN interactions: chromatin, nuclear pore, ER membrane and axonemal proteins. To capture co-variation of bound proteins between different experiments (comparing GFP-only with SUN1-GFP and ALLAN-GFP), we performed a principal component analysis (PCA) using unique peptide values per protein present in two replicates per experiment. Values for undetected proteins were set to 0. Values were ln (x + 1) transformed and PCA was performed using the ClustVis webserver (settings Nipals PCA, no scaling) (*Metsalu and Vilo, 2015*).

## AlphaFold3 modelling

3D protein structures for SUN1 and ALLAN were modelled using the AlphaFold3 webserver (https://alphafoldserver.com/) with standard settings (seed set to 100) (*Abramson et al., 2024*). To assess

the stoichiometry of the complex we modelled from monomers up to decamers for both ALLAN and SUN1, using the N-terminus, the middle domain and the C-terminus of SUN1. ALLAN consistently formed higher order structures containing trimeric complexes. Although SUN1 could form higher order structures beyond trimers, we opted to model it as a trimer given the preference of ALLAN to form trimers.

## Fixed Immunofluorescence assay and DNA content analysis

SUN1-KO and ALLAN-KO gametocytes were purified, activated in ookinete medium, fixed at various time points with 4% paraformaldehyde (PFA, Sigma) diluted in microtubule (MT)-stabilising buffer (MTSB) for 10 to 15 min, and added to poly-L-lysine coated slides. Immunocytochemistry used mouse anti-α tubulin mAb (Sigma-T9026; used at 1:1000) as primary antibody, and secondary antibody was Alexa 568 conjugated anti-mouse IgG (Invitrogen-A11004) (used at 1:1000). Slides were mounted in Vectashield with DAPI (Vector Labs) for fluorescence microscopy with a Zeiss AxioImager M2 microscope fitted with an AxioCam ICc1 digital camera.

To measure nuclear DNA content of activated male gametocytes by direct immunofluorescence, images of parasites fixed (8 min) and stained as above were analysed using the ImageJ software (version 1.44) (National Institute of Health). Z-stack images were taken and a sum slice projection produced. The nuclear boundary was drawn around DNA staining using the freehand tool and measurements were taken from the DAPI channel including area, mean intensity and integrated density. The same measurements were also taken for an area of the background surrounding the cell. The ring stage of parasite was used as control (1N). A value for corrected intensity was calculated using the formula: corrected intensity = integrated density of nucleus – (area of nucleus*mean of background).

## Liver stage cultures

Human liver hepatocellular carcinoma (HepG2) cells (European Cell Culture Collection) were grown in DMEM medium, supplemented with 10% heat inactivated FBS, 0.2% $NaHCO_3$, 1% sodium pyruvate, 1% penicillin/streptomycin and 1% L-glutamine in a humidified incubator at 37°C with 5% $CO_2$. For infection, $1 \times 10^5$ HepG2 cells were seeded in a 48-well culture plate. The day after seeding, sporozoites were purified following mechanical disruption from salivary glands removed from female *A. stephensi* mosquitoes infected with PbALLAN-GFP parasites, and added in infection medium to the culture maintained for 72 hr.

## Ultrastructure expansion microscopy (UExM)

Purified gametocytes were activated for different time periods, and then activation was stopped by adding 4% paraformaldehyde. Fixed cells were then attached to a 12 mm diameter poly-D-lysine (A3890401, Gibco) coated coverslip for 10 min. Coverslips were incubated overnight in 1.4% formaldehyde (FA)/2% acrylamide (AA) at 4°C. Gelation was performed in ammonium persulphate (APS)/ TEMED (10% each)/monomer solution (23% sodium acrylate; 10% AA; 0.1% BIS-AA in PBS) for 1 hr at 37°C. Gels were denatured for 15 min at 37°C and 45 min at 95 °C and then incubated in distilled water overnight for complete expansion. The following day, gels were washed twice in PBS for 15 min to remove excess water, and then incubated with primary antibodies at 37 °C for 3 hr. After washing three times for 10 min in PBS/ 0.1% Tween, incubation with secondary antibodies was performed for 3 hr at 37°C followed by three washes of 10min each in PBS/ 0.1% Tween (all antibody incubation steps were performed at 37 °C with 120 to 160 rpm shaking). Directly after antibody staining, gels were incubated in 1 ml of 594 NHS-ester (Merck: 08741) diluted to 10 µg/ml in PBS for 90 min at room temperature on a shaker. The gels were then washed three times for 15 min with PBS/0.1% Tween and expanded overnight in ultrapure water. One cm × 1 cm pieces were cut from the expanded gel and attached to 24 mm diameter Poly-D-Lysine (A3890401, Gibco) coated coverslips to prevent the gel from sliding and avoid drifting while imaging. The primary antibody was either against α-tubulin (1:1000 dilution, Sigma-T9026), or an anti-GFP antibody (1:250, Thermo Fisher). Secondary antibodies were anti-rabbit Alexa 488, or anti-mouse Alexa 568 (Invitrogen), used at dilutions of 1:1000. Atto 594 NHS-ester (Merck 08741) was used for bulk proteome labelling. Images were acquired on Zeiss Elyra PS.1-LSM780 and CD7-LSM900, and Airyscan confocal microscopes, where 0.4 Airy unit (AU) on confocal and 0.2 AU were used along with slow scan modes. Image analysis was performed using Zeiss Zen 2012 Black edition and Fiji-Image J.

## Structured illumination microscopy

A small volume (3 µl) of gametocyte suspension was placed on a microscope slide and covered with a long (50 × 34 mm) coverslip to obtain a very thin monolayer and immobilise the cells. Cells were scanned with an inverted microscope using a Zeiss Plan-Apochromat 63x/1.4 oil immersion or Zeiss C-Apochromat 63x/1.2 W Korr M27 water immersion objective on a Zeiss Elyra PS.1 microscope, utilising the structured illumination microscopy (SIM) technique. The correction collar of the objective was set to 0.17 for optimal contrast. The following settings were used in SIM mode: lasers, 405 nm: 20%, 488 nm: 16%; exposure times 200 ms (Hoechst), 100 ms (GFP), three grid rotations, five phases. The bandpass filters BP 420-480 + LP 750, and BP 495-550+LP 750 were used for the blue and green channels, respectively. Where multiple focal planes (Z-stacks) were recorded processing and channel alignment was done as described previously (*Zeeshan et al., 2024*).

## Electron microscopy

Gametocytes activated for 8 min and 15 min were fixed in 4% glutaraldehyde in 0.1 M phosphate buffer and processed for electron microscopy (*Ferguson et al., 2005*). Briefly, samples were post-fixed in osmium tetroxide, treated en bloc with uranyl acetate, dehydrated, and embedded in Spurr's epoxy resin. Thin sections were stained with uranyl acetate and lead citrate prior to examination in a JEOL JEM-1400 electron microscope (JEOL, United Kingdom). The experiments were done at least three times to capture every stage of the parasite, and 50 to 55 cells were examined for phenotypic analysis. For more details, please see the figure legends.

## RNA isolation and quantitative real-time PCR (qRT-PCR) analyses

RNA was extracted from purified gametocytes using an RNA purification kit (Stratagene). Complementary DNA (cDNA) was synthesised using an RNA-to-cDNA kit (Applied Biosystems). Gene expression was quantified from 80 ng of total RNA using the SYBR Green Fast Master Mix kit (Applied Biosystems). All primers were designed using Primer3 (Primer-BLAST, NCBI). The analysis was conducted on an Applied Biosystems 7500 Fast machine with the following cycling conditions: 95°C for 20 s, followed by 40 cycles of 95°C for 3 s and 60°C for 30 s. Three technical replicates and three biological replicates were performed for each gene tested. The genes hsp70 (PBANKA_081890) and arginyl-tRNA synthetase (PBANKA_143420) were used as endogenous control reference genes. The primers used for qPCR are listed in *Table 1*.

## RNA-seq analysis

Libraries were prepared from lyophilised total RNA, starting with the isolation of mRNA using the NEBNext Poly(A) mRNA Magnetic Isolation Module (NEB), followed by the NEBNext Ultra Directional RNA Library Prep Kit (NEB) as per the manufacturer's instructions. Libraries were amplified through 12 PCR cycles (12 cycles of [15 s at 98°C, 30 s at 55°C, 30 s at 62°C]) using the KAPA HiFi HotStart Ready Mix (KAPA Biosystems). Sequencing was performed on a NovaSeq 6000 DNA sequerncer (Illumina), generating paired-end 100-bp reads.

FastQC (https://www.bioinformatics.babraham.ac.uk/projects/fastqc/; *Andrews, 2023*) was used to analyse raw read quality and the adapter sequences were removed using Trimmomatic (v0.39) (http://www.usadellab.org/cms/?page=trimmomatic; *Bolger et al., 2018*). The resulting reads were mapped against the *P. berghei* genome (PlasmoDB, v68) using HISAT2 (v2-2.2.1) with the `--very-sensitive parameter`. Uniquely mapped, properly paired reads with mapping quality of 40 or higher were retained using SAMtools (v1.19) (http://samtools.sourceforge.net/; *Li, 2018*). Raw read counts were determined for each gene in the *P. berghei* genome using BedTools (https://bedtools.readthedocs.io/en/latest/#; *Quinlan, 2023*) to intersect the aligned reads with the genome annotation. Differential expression analysis was performed using DESeq2 to call up-and down-regulated genes (FDR < 0.05 and log2 FC > 1.0). Volcano plots were made using the R package Enhanced Volcano.

## Lipidomics analyses

Lipidomics analysis was performed on 3-4 independent cell harvests of the Δ*sun1* and WT-GFP gametocytes. Gametocytes were purified as described above and metabolically quenched at 0 min

and 8 min post activation via rapid cooling to 0°C by suspending the tube over a dry ice/100% ethanol slurry mix, while continually stirring the solution.

After three washing steps with 1 ml ice-cold PBS followed by centrifugation, the parasite pellet was subjected to lipid extraction using chloroform and methanol in the presence of butylhydroxytoluene, PC (C21:0/C21:0), C13:0 or C15:0 FA standards. Lipid extraction, separation, and analyses were all performed as previously described (*Flammersfeld et al., 2020*). Briefly, dried down samples were reconstituted in 80 µL methanol and incubated at 30°C for 5 min with vigorous vortex. Samples (1 µL) were analysed by LCMS (Agilent 1290 infinity/Infinity II Agilent) MS (Agilent 6495c triple quadruple). Acquisition DMRM (dynamic multiple reaction monitoring) method was used as described previously (*Charital et al., 2024*) with a modification for *Plasmodium* lipid species. The LCMS data was subjected to targeted analysis using Mass Hunter Quantification software (Agilent). Each lipid species was quantified using a calibration curve of each representative lipid with known abundance. Then each lipid abundance was normalised according to the cell ratio.

The graphical data for this study was generated using GraphPad Prism software. Three biological replicates were used per experiment (at least n = 3, unless stated otherwise). The error bars are representative of the standard error of mean (SEM) for each study. Statistical significance was determined for each experiment by unpaired t tests using GraphPad Prism. The range of statistical significance was signified as per the *p* value indicated in each graph.

## Statistical analysis

All statistical analyses were conducted using GraphPad Prism 9 (GraphPad Software). Student's t-test and/or a two-way ANOVA test were employed to assess differences between control and experimental groups. Statistical significance is indicated as *$p<0.05$, **$p<0.01$, **$p<0.001$, or ns for not significant. 'n' denotes the sample size in each group or the number of biological replicates. For qRT-PCR data, a multiple comparisons t-test, with post hoc Holm–Sidak test, was utilised to evaluate significant differences between wild-type and mutant parasites.

## Materials availability

Newly created parasite lines in this study are available from Nottingham laboratory and can be accessed upon request from corresponding author Prof. Rita Tewari.

## Acknowledgements

RT is supported by ERC advance grant funded by UKRI Frontier Science (EP/X024776/1), MRC UK (MR/K011782/1), and BBSRC (BB/L013827/1, BB/X014681/1). MZ, IB., DB AM. were supported as research fellows by (EP/X024776/1). RY is supported by BBSRC (BB/X014681/1). SLP is supported by Wellcome DBT India Alliance/Team Science (IA/TSG/21/1/600261). AAH is supported by the Francis Crick Institute (FC001097), which receives core funding from the Cancer Research UK (FC001097), the UK Medical Research Council (FC001097), and the Wellcome Trust (FC001097). KGLR is supported by the NIH/NIAID (R01 AI136511) and the University of California, Riverside NIFA-Hatch-225935. ET was supported by a personal fellowship from the Nederlandse Organisatie voor Wetenschappelijk Onderzoek (NOW), the Netherlands (grant no. VI. Veni.202.223). YYB and CYB were supported by Agence Nationale de la Recherche, France (Project ApicoLipiAdapt grant ANR-21-CE44-0010; Project Apicolipidtraffic grant ANR-23-CE15-0009-01; Project OIL grant ANR-24-CE15-2171-02), The Fondation pour la Recherche Médicale (FRM EQU202103012700), Laboratoire d'Excellence Parafrap, France (grant ANR-11-LABX-0024), LIA-IRP CNRS Program (Apicolipid project), the Université Grenoble Alpes (IDEX ISP Apicolipid) and Région Auvergne Rhone-Alpes for the lipidomics analyses platform (Grant IRICE Project GEMELI), Collaborative Research Program Grant CEFIPRA (Project 6003–1) by the CEFIPRA (MESRI-DBT). Confocal and SIM microscopy was conducted in the School of Life Sciences Imaging (SLIM). For Open Access, the authors have applied a CC BY public copyright licence to any Author Accepted Manuscript version arising from this submission. We thank Cleidiane Zampronio at Warwick University for mass spectrometry methods and Bio Support Unit, University of Nottingham for maintenance of mice used in this study.

## Additional information

### Funding

| Funder | Grant reference number | Author |
|---|---|---|
| European Research Council | EP/X024776/1 | Mohammad Zeeshan<br>Igor Blatov<br>Akancha Mishra<br>Declan Brady<br>Rita Tewari |
| Medical Research Council | MR/K011782/1 | Rita Tewari |
| Biotechnology and Biological Sciences Research Council | BB/L013827/1 | Rita Tewari |
| Biotechnology and Biological Sciences Research Council | BB/X014681/1 | Ryuji Yanase<br>Rita Tewari |
| Wellcome Trust DBt India Alliance | Team Science IA/TSG/21/1/600261 | Sarah L Pashley<br>Rita Tewari |
| Francis Crick Institute | FC001097 | Anthony A Holder |
| National Institute of Allergy and Infectious Diseases | R01 AI136511 | Karine G Le Roch |
| Agence Nationale de la Recherche | ANR-21-CE44-0010 | Yoshiki Yamaryo-Botté<br>Cyrille Y Botté |
| Agence Nationale de la Recherche | Project Apicolipidtraffic ANR-23-CE15-0009-01 | Yoshiki Yamaryo-Botté<br>Cyrille Y Botté |
| Agence Nationale de la Recherche | Project OIL ANR-24-CE15-2171-02 | Yoshiki Yamaryo-Botté<br>Cyrille Y Botté |
| Fondation pour la Recherche Médicale | FRM EQU202103012700 | Yoshiki Yamaryo-Botté<br>Cyrille Y Botté |
| Laboratoire d'Excellence Parafrap | ANR-11-LABX-0024 | Yoshiki Yamaryo-Botté<br>Cyrille Y Botté |
| Centre Franco-Indien pour la Promotion de la Recherche Avancée | Collaborative Research Program Grant Project 6003–1 | Yoshiki Yamaryo-Botté<br>Cyrille Y Botté |
| Nederlandse Organisatie voor Wetenschappelijk Onderzoek | Veni.202.223 | Eelco Tromer |
| University of California, Riverside | NIFA-Hatch-225935 | Karine G Le Roch |

The funders had no role in study design, data collection and interpretation, or the decision to submit the work for publication. For the purpose of Open Access, the authors have applied a CC BY public copyright license to any Author Accepted Manuscript version arising from this submission.

### Author contributions

Mohammad Zeeshan, Formal analysis, Validation, Investigation, Visualization, Methodology, Writing – original draft, Writing – review and editing; Igor Blatov, Ryuji Yanase, Sarah L Pashley, Validation, Investigation, Visualization, Methodology; David JP Ferguson, Formal analysis, Supervision, Validation, Investigation, Visualization, Methodology, Writing – original draft, Writing – review and editing; Zeinab Chahine, Baptiste Marche, Suhani Bhanvadia, Molly Hair, Sagar Batra, Investigation, Methodology; Yoshiki Yamaryo-Botté, Declan Brady, Formal analysis, Investigation, Methodology; Akancha Mishra, Investigation, Visualization, Methodology; Robert Markus, Software, Visualization; Andrew R Bottrill, Software, Investigation, Methodology; Sue Vaughan, Software, Methodology; Cyrille Y Botté, Formal analysis, Funding acquisition, Investigation, Methodology; Karine G Le Roch, Software, Funding acquisition; Anthony A Holder, Supervision, Funding acquisition, Writing – review and editing;

Eelco Tromer, Software, Funding acquisition, Methodology, Writing – original draft, Writing – review and editing; Rita Tewari, Conceptualization, Resources, Data curation, Formal analysis, Supervision, Funding acquisition, Investigation, Visualization, Methodology, Writing – original draft, Project administration, Writing – review and editing

## Author ORCIDs
Mohammad Zeeshan ⓘ https://orcid.org/0000-0002-6185-403X
Igor Blatov ⓘ https://orcid.org/0000-0003-2110-0441
Ryuji Yanase ⓘ https://orcid.org/0000-0001-9419-398X
David JP Ferguson ⓘ https://orcid.org/0000-0001-5045-819X
Zeinab Chahine ⓘ https://orcid.org/0000-0001-8208-5869
Robert Markus ⓘ https://orcid.org/0000-0003-4535-303X
Declan Brady ⓘ https://orcid.org/0000-0001-9150-7840
Andrew R Bottrill ⓘ https://orcid.org/0000-0002-5182-3643
Cyrille Y Botté ⓘ https://orcid.org/0000-0002-2245-536X
Karine G Le Roch ⓘ https://orcid.org/0000-0002-4862-9292
Anthony A Holder ⓘ https://orcid.org/0000-0002-8490-6058
Eelco Tromer ⓘ https://orcid.org/0000-0003-3540-7727
Rita Tewari ⓘ https://orcid.org/0000-0003-3943-1847

## Ethics
The animal work passed an ethical review process and was approved by the United Kingdom Home Office. Work was carried out under UK Home Office Project Licenses (PDD2D5182 and PP3589958) in accordance with the UK 'Animals (Scientific Procedures) Act 1986'. Six- to eight-week-old female CD1 outbred mice from Charles River Laboratories were used for all experiments. The mice were maintained under a 12 hr light and 12 hr dark (7 am till 7 pm) cycle, at a temperature between 20 and 24 °C, and a humidity between 40 and 60%.

Reviewer #1 (Public review): https://doi.org/10.7554/eLife.106537.2.sa1
Reviewer #2 (Public review): https://doi.org/10.7554/eLife.106537.2.sa2
Author response https://doi.org/10.7554/eLife.106537.2.sa3

# Additional files

## Supplementary files
MDAR checklist

## Data availability
The RNA-seq data generated in this study have been deposited in the NCBI Sequence Read Archive with accession number PRJNA1246157. Mass spectrometry proteomics data have been deposited to the ProteomeXchange Consortium via the PRIDE partner repository and data are available with identifier PXD062729. Source data are provided with this paper.

The following datasets were generated:

| Author(s) | Year | Dataset title | Dataset URL | Database and Identifier |
| --- | --- | --- | --- | --- |
| Zeeshan et al. | 2025 | A novel SUN1-ALLAN complex coordinates segregation of the bipartite MTOC across the nuclear envelope during rapid closed mitosis in Plasmodium | https://www.ncbi.nlm.nih.gov/sra/?term=PRJNA1246157 | NCBI Sequence Read Archive, PRJNA1246157 |

*Continued on next page*

*Continued*

| Author(s) | Year | Dataset title | Dataset URL | Database and Identifier |
|---|---|---|---|---|
| Bottrill A, Tewari R | 2025 | A novel SUN1-ALLAN complex coordinates segregation of the bipartite MTOC across the nuclear envelope during rapid closed mitosis in Plasmodium | https://www.ebi.ac.uk/pride/archive/projects/PXD062729 | PRIDE, PXD062729 |

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
