## [Editor Report · eLife Assessment]

In Plasmodium male gametocytes, rapid nuclear division occurs with an intact nuclear envelope, requiring precise coordination between nuclear and cytoplasmic events to ensure proper packaging of each nucleus into a developing gamete. This **valuable** study characterizes two proteins involved in the formation of *Plasmodium berghei* male gametes. By integrating live-cell imaging, ultrastructural expansion microscopy, and proteomics, this study **convincingly** identifies SUN1 and its interaction partner ALLAN as crucial nuclear envelope components in male gametogenesis. A role for SUN1 in membrane dynamics and lipid metabolism is less well supported. The results are of interest for general cell biologists working on unusual mitosis pathways.

[Editors' note: this paper was reviewed by Review Commons.]

---

## [Referee Report · Reviewer #1 (Public review)]

Summary:

Activated male Plasmodium gametocytes undergo very rapid nuclear division, while keeping the nuclear envelope intact. There is interest in how events inside the nucleus are co-ordinated with events in the parasite cytoplasm, to ensure that each nucleus is packaged into a nascent male gamete.

This manuscript by Zeeshan et al describes the organisation of a nuclear membrane bridging protein, SUN1, during nuclear division. SUN1 is expected from studies in other organisms to be a component of a bridging complex (LINC) that connects the inner nuclear membrane to the outer nuclear membrane, and from there to the cytoplasmic microtubule-organising centres, the centrosome and the basal body.

The authors show that knockout of the SUN1 in gametocytes leads to severe disruption of the mitotic spindle and failure of the basal bodies to segregate. The authors show convincingly that functional SUN1 is required for male gamete formation and subsequent oocyst development.

The authors identified several SUN1-interacting proteins, thus providing information about the nuclear membrane bridging machinery.

Strengths:

The authors have used state of the art imaging, genetic manipulation and immunoprecipitation approaches.

Weaknesses:

Technical limitations of some of the methods used make it difficult to interpret some of the micrographs.

From studies in other organisms, a protein called KASH is a critical component the bridging complex (LINC). That is, KASH links SUN1 to the outer nuclear membrane. The authors undertook a gene sequence analysis that reveals that Plasmodium lacks a KASH homologue. Thus, further work is needed to identify the functional equivalent of KASH, to understand bridging machinery in Plasmodium.

Comments on revised version:

The authors have addressed the comments and suggestions that I provided as part of a Review Commons assessment.

---

## [Referee Report · Reviewer #2 (Public review)]

Zeeshan et al. investigate the function of the protein SUN1, a proposed nuclear envelope protein linking nuclear and cytoplasmic cytoskeleton, during the rapid male gametogenesis of the rodent malaria parasite *Plasmodium berghei*. They reveal that SUN1 localises to the nuclear envelope (NE) in male and female gametes and show that the male NE has unexpectedly high dynamics during the rapid process of gametogenesis. Using expansion microscopy, the authors find that SUN1 is enriched at the neck of the bipartite MTOC that links the intranuclear spindle to the basal bodies of the cytoplasmic axonemes. Upon deletion of SUN1, the basal bodies of the eight axonemes fail to segregate, no spindle is formed, and emerging gametes are anucleated, leading to a complete block in transmission. By interactomics the authors identify a divergent allantoicase-like protein, ALLAN, as a main interaction partner of SUN1 and further show that ALLAN deletion largely phenocopies the effect of SUN1.

Overall, the authors use an extensive array of fluorescence and electron microscopy techniques as well as interactomics to convincingly demonstrate that SUN1 and ALLAN play a role in maintaining the structural integrity of the bipartite MTOC during the rapid rounds of endomitosis in male gametogenesis.

Two suggestions for improvement of the work remain:

(1) Lipidomic analysis of WT and SUN1-knockout gametocytes before and after activation resulted in only minor changes in some lipid species. Without statistical analysis, it remains unclear if these changes are statistically significant and not rather due to expected biological variability. While the authors clearly toned down their conclusions in the revised manuscript, some phrasings in the results and the discussion still suggest that gametocyte activation and/or SUN1-knockout affects lipid composition. Similarly, some phrases suggest that SUN1 is responsible for the observed loops and folds in the NE and that SUN1 KO affects the NE dynamics. Currently, I do not think that the data supports these statements.

(2) It is interesting to note that ALLAN has a much more specific localisation to basal bodies than SUN1, which is located to the entire nuclear envelope. Knock out of ALLAN also exhibits a milder (but still striking) phenotype than knockout of SUN1. These observations suggest that SUN1 has additional roles in male gametogenesis besides its interaction with ALLAN, which could be discussed a bit more.

This study uses extensive microscopy and genetics to characterise an unusual SUN1-ALLAN complex, thus providing new insights into the molecular events during Plasmodium male gametogenesis, especially how the intranuclear events (spindle formation and mitosis) are linked to the cytoplasmic separation of the axonemes. The characterisation of the mutants reveals an interesting phenotype, showing that SUN1 and ALLAN are localised to and maintain the neck region of the bipartite MTOC. The authors here confirm and expand the previous knowledge about SUN1 in P. berghei, adding more detail to its localisation and dynamics, and further characterise the interaction partner ALLAN. Given the evolutionary divergence of Plasmodium, these results are interesting not only for parasitologists, but also for more general cell biologists.

---

## [Author Response]

**Reviewer #1 (Evidence, reproducibility and clarity):**
Minor comments:In the results section (lines 498-499), the authors describe free kinetochores in many cells without associated spindle microtubules. However, some nuclei appear to have kinetochores, as presented in Figure 6. Could the authors clarify how this conclusion was derived using transmission electron microscopy (TEM) without serial sectioning, as this is not explicitly mentioned in the materials and methods?

We observed free kinetochores in the ALLAN-KO parasites with no associated spindle microtubules (see Fig. 6Gh), while kinetochores are attached to spindle microtubules in WT-GFP cells (see Fig. 6Gc). To provide further evidence we analysed additional images and found that ALLAN-KO cells have free kinetochores in the centre of nucleus, unattached to spindle microtubules. We provide some more images clearly showing free kinetochores in these cells (new supplementary Fig. S11).

However, in the ALLAN mutant, this difference is not absolute: in a search of over 50 cells, one example of a cell with a “normal” nuclear spindle and attached kinetochores was observed.

The use of serial sectioning has limitations for examining small structures like kinetochores in whole cells. The limitations of the various techniques (for example, SBF-SEM vs tomography) are highlighted in our previous study (Hair et al 2022; PMID: 38092766), and we consider that examining a population of randomly sectioned cells provides a better understanding of the overall incidence of specific features.

Discussion Section:Could the authors expand on why SUN1 and ALLAN are not required during asexual replication, even though they play essential roles during male gametogenesis?

We observed no phenotype in asexual blood stage parasites associated with the *sun1* and *allan* gene deletions. Several other *Plasmodium berghei* gene knockout parasites with a phenotype in sexual stages, for example CDPK4 (PMID: 15137943), SRPK (PMID: 20951971), PPKL (PMID: 23028336) and kinesin-5 (PMID: 33154955) have no phenotype in blood stages, so perhaps this is not surprising. One explanation may be the substantial differences in the mode of cell division between these two stages. Asexual blood stages produce new progeny (merozoites) over 24 hours with closed mitosis and asynchronous karyokinesis during schizogony, while male gametogenesis is a rapid process, completed within 15 min to produce eight flagellated gametes. During male gametogenesis the nuclear envelope must expand to accommodate the increased DNA content (from 1N to 8N) before cytokinesis. Furthermore, male gametogenesis is the only stage of the life cycle to make flagella, and axonemes must be assembled in the cytoplasm to produce the flagellated motile male gametes at the end of the process. Thus, these two stages of parasite development have some very different and specific features.

Lines 611-613 states: "These loops serve as structural hubs for spindle assembly and kinetochore attachment at the nuclear MTOC, separating nuclear and cytoplasmic compartments." Could the authors elaborate on the evidence supporting this statement?

We observed the loops/folds in the nuclear envelope (NE) as revealed by SUN1-GFP and 3D TEM images during male gametogenesis. These folds/loops occur mainly in the vicinity of the nuclear MTOC where the spindles are assembled (as visualised by EB1 fluorescence) and attached to kinetochores (as visualised by NDC80 fluorescence). These loops/folds may form due to the contraction of the spindle pole back to the nuclear periphery, inducing distortion of the NE. Since there is no physical segregation of chromosomes during the three rounds of mitosis (DNA increasing from 1N to 8N), we suggest that these folds provide additional space for spindle and kinetochore dynamics within an intact NE to maintain separation from the cytoplasm (as shown by location of kinesin-8B).

In lines 621-622, the authors suggest that ALLAN may have a broader role in NE remodelling across the parasite's lifecycle. Could they reflect on or remind readers of the finding that ALLAN is not essential during the asexual stage?

ALLAN-GFP is expressed throughout the parasite life cycle but as the reviewer points out, a functional role is more pronounced during male gametogenesis. This does not mean that it has no role at other stages of the life cycle even if there is no obvious phenotype following deletion of the gene during the asexual blood stage. The fact that ALLAN is not essential during the asexual blood stage is noted in lines 628-29.

**Reviewer #2 (Evidence, reproducibility and clarity):**
IntroductionLine 63: The authors stat: "NE is integral to mitosis, supporting spindle formation, kinetochore attachment, and chromosome segregation..". Seemingly at odds, they also say (Line 69) that 'open' "mitosis is "characterized by complete NE disassembly".The authors could explain better the ideas presented in their quoted review from Dey and Baum, which points out that truly 'open' and 'closed' topologies may not exist and that even in 'open' mitosis, remnants of the NE may help support the mitotic spindle.

We have modified the sentence in which we discuss current opinions about ‘open’ and ‘closed’ mitosis. It is believed that there is no complete disassembly of the NE during open mitosis and no completely intact NE during closed mitosis, respectively. In fact, the NE plays a critical role in the different modes of mitosis during MTOC organisation and spindle dynamics. Please see the modified lines 64-71.

ResultsFig 7 is the final figure; but would be more useful upfront.

We have provided a new introductory figure (Fig 1) showing a schematic of conventional /canonical LINC complexes and evidence of SUN protein functions in model eukaryotes and compare them to what is known in apicomplexans.

Fig 1D. The authors generated a C-terminal GFP-tagged SUN1 transfectants and used ultrastructure expansion microscopy (U-ExM) and structured illumination microscopy (SIM) to examine SUN1-GFP in male gametocytes post-activation. The immuno-labelling of SUN1-GFP in these fixed cells appears very different to the live cell images of SUN1-GFP. The labelling profile comprises distinct punctate structures (particularly in the U-ExM images), suggesting that paraformaldehyde fixation process, followed by the addition of the primary and secondary antibodies has caused coalescing of the SUN1-GFP signal into particular regions within the NE.

We agree with the reviewer. Fixation with paraformaldehyde (PFA) results in a coalescence of the SUN1-GFP signal. We have also tried methanol fixation (see new Fig. S2), but a similar problem was encountered.

Given these fixation issues, the suggestion that the SUN1-GFP signal is concentrated at the BB/ nuclear MTOC and "enriched near spindle poles" needs further support.These statements seem at odd with the data for live cell imaging where the SUN1-GFP seems evenly distributed around the nuclear periphery. Can the observation be quantitated by calculating the percentage of BB/ nuclear MTOC structures with associated SUN1-GFP puncta? If not, I am not convinced these data help understand the molecular events.

We agree with the reviewer that whilst the live cell imaging showed an even distribution of SUN1-GFP signal, after fixation with either PFA or methanol, then SUN1-GFP puncta are observed in addition to the peripheral location around the stained DNA (Hoechst) (See Fig. S2; puncta are indicated by arrows). These SUN1-GFP labelled puncta were observed at the junction of the nuclear MTOC and the basal body (Fig. 2F). Quantification of the distribution showed that these SUN1-GFP puncta are associated with nuclear MTOC in more than 90 % of cells (18 cells examined). Live cell imaging of the dual labelled parasites; SUN1xkinesin-8B (Fig. 2H) and SUN1x EB1 (Fig. 2I) provides further support for the association of SUN1-GFP puncta with BB (kinesin-8B) /nuclear MTOC (EB1).

The authors then generated dual transfectants and examined the relative locations of different markers in live cells. These data are more informative.The authors state; " ..SUN1-GFP marked the NE with strong signals located near the nuclear MTOCs situated between the BB tetrads". The nuclear MTOCs are not labelled in this experiment. The SUN1-GFP signal between the kinesin-8B puncta is evident as small puncta on regions of NE distortion. I would prefer to not describe this signal as "strong". The signal is stronger in other regions of the NE.

We have modified the sentence on line 213 to accommodate this suggestion.

Line 219. The authors state; "..SUN1-GFP is partially colocalized with spindle poles as indicated by EB1,.. it shows no overlap with kinetochores (NDC80)." The authors should provide an analysis of the level of overlap at a pixel by pixel level to support this statement.

We now provide the overlap at a pixel-by-pixel level for representative images, and we have quantified more cells (n>30), as documented in the new Fig. S4A. We have also modified the sentence on line 219 to reflect these additions.

The SUN1 construct is C-terminally GFP-tagged. By analogy with human SUN1, the C-terminal SUN domain is expected to be in the NE lumen. That is in a different compartment to EB1, which is located in the nuclear lumen (on the spindle). Thus, the overlap of signal is expected to be minimal.

We agree with the reviewer that the overlap between EB1 and Sun1 signals is expected to be minimal. We have quantified the data and included it in Supplementary Fig. S4A.

Similarly, given that EB1 and NDC80 are known to occupy overlapping locations on the spindle, it seems unlikely that SUN1 can overlap with one and not the other.

We agree with the reviewer’s analysis that EB1 and NDC80 occupy overlapping locations on the spindle, although the length of NDC80 is less at the ends of spindles (see Author response image 1A) as shown in our previous study where we compared the locations of two spindle proteins, ARK2 and EB1, with that of NDC80 (Zeeshan et al, 2022; PMID: 37704606). In the present study we observed that Sun1-GFP partially overlaps with EB1 at the ends of the spindle, but not with NDC80. Please see Author response image 1B.

I note on Line 609, the authors state "Our study demonstrates that SUN1 is primarily localized to the nuclear side of the NE.." As per Fig 7D, and as discussed above, the bulk of the protein, including the SUN1 domain, is located in the space between the INM and the ONM.

We appreciate the reviewer’s correction; we have now modified the sentence to indicate that the protein is largely localized in the space between the INM and the ONM on line 617.

Interestingly, as the authors point out, nuclear membrane loops are evident around EB1 and NDC80 focal regions. The data suggests that the contraction of the spindle pole back to the nuclear periphery induces distortion of the NE.

We agree with the reviewer’s suggestion that the data indicate that contraction of spindle poles back to the nuclear periphery may induce distortion of the NE.

The author should discuss further the overlap of findings of this study with that from a recent manuscript (https://doi.org/10.1016/j.cels.2024.10.008). That Sayers et al. study identified a complex of SUN1 and ALLC1 as essential for male fertility in P. berghei. Sayers et al. also provide evidence that this complex particulate in the linkage of the MTOC to the NE and is needed for correct mitotic spindle formation during male gametogenesis.

We thank the reviewer for this suggestion. The study by Sayers et al, (2024) was published while our manuscript was under preparation. It was interesting to see that these complementary studies have similar findings about the role of SUN1 and the novel complex of SUN1-ALLAN. Our study contains a more detailed, in-depth analysis both by Expansion and TEM of SUN1. We include additional studies on the role of ALLAN. We discuss the overlap in the findings of the two studies in lines 590-605.

While the work is interesting, the conclusions may need to be tempered. The authors suggestion that in the absence of KASH-domain proteins, the SUN1-ALLAN complex forms a non-canonical LINC complex (that is, a connection across the NE), that "achieves precise nuclear and cytoskeletal coordination".

We have toned down the wording of this conclusion in lines 665-677.

In other organisms, KASH interacts with the C-terminal domain on SUN1, which as mentioned above is located between the INM and ONM. By contrast, ALLAN interacts with the N-terminal domain of SUN1, which is located in the nuclear lumen. The SUN1-ALLAN interaction is clearly of interest, and ALLAN might replace some of the roles of lamins. However, the protein that functionally replaces KASH (i.e. links SUN1 to the ONM) remains unidentified.

We agree with reviewer, and future studies will need to focus on identifying the KASH replacement that links SUN1 to the ONM.

It may also be premature to suggest that the SUN1-ALLAN complex is promising target for blocking malaria transmission. How would it be targeted?

We have deleted the sentence that raised this suggestion.

While the above datasets are interesting and internally consistent, there are two other aspects of the manuscript that need further development before they can usefully contribute to the molecular story.The authors undertook a transcriptomic analysis of Δsun1 and WT gametocytes, at 8 and 30 min post-activation, revealing moderate changes (~2-fold change) in different genes. GO-based analysis suggested up-regulation of genes involved in lipid metabolism. Given the modest changes, it may not be correct to conclude that "lipid metabolism and microtubule function may be critical functions for gametogenesis that can be perturbed by sun1 deletion." These changes may simply be a consequence of the stalled male gametocyte development.

Following the reviewer’s suggestion we have moved these data to the supplementary information (Fig. S5D-I) and toned down their discussion in the results and discussion sections.

The authors have then undertaken a detailed lipid analysis of the Δsun1 and WT gametocytes, before and after activation. Substantial changes in lipid metabolites might not be expected in such a short period of time. And indeed, the changes appear minimal. Similarly, there are only minor changes in a few lipid sub-classes between Δsun1 and WT gametocytes. In my opinion, the data are not sufficient to support the authors conclusion that "SUN1 plays a crucial role, linking lipid metabolism to NE remodelling and gamete formation."

In agreement with the reviewer’s comments we have moved these data to supplementary information (Fig. S6) and substantially toned down the conclusions based on these findings.

**Reviewer #3 (Evidence, reproducibility and clarity):**
Major comments:My main concern with this manuscript is that the authors do conclude not only that SUN1 is important for spindle formation and basal body segregation, but also that it influences for lipid metabolism and NE dynamics. I don't think the data supports this conclusion, for several reasons listed below. I would suggest to remove this claim from the manuscript or at least tone it down unless more supporting data are provided, in particular showing any change in NE dynamics in the SUN1-KO. Instead I would recommend to focus on the more interesting role of SUN1-ALLAN in bipartite MTOC organisation, which likely explains all observed phenotypes (including those in later stages of the parasite life cycle). In addition, some aspects of the knockout phenotype should be quantified to a bit deeper level.In more detail:- The lipidomics analysis is clearly the weakest point of the manuscript: The authors state that there are significant changes in some lipid populations between WT and sun1-KO, and between activated and non-activated cells, yet no statistical analysis is shown and the error bars are quite high compared to only minor changes in the means. For some discussed lipids, the result text does not match the graphs, e.g. PA, where the increase upon activation is more pronounced in the SUN1-KO vs WT (contrary to the text), or MAG, which is reduced in the SUN1-KO vs WT (contrary to the text). I don't see the discussed changes in arachidonic acid levels and myristic acid levels in the data either. Even if the authors find after analysis some statistically significant differences between some groups, they should carefully discuss the biological significance of these differences. As it is, I do not think the presented data warrants the conclusion that deletion of SUN1 changes lipid homeostasis, but rather shows that overall lipid homeostasis is not majorly affected by gametogenesis or SUN1 deletion. As a minor comment, if you decide to keep the lipidomics analysis in the manuscript, please state how many replicates were done.

As detailed above we have moved the lipidomics data to supplementary information (Fig. S6) and substantially toned down the discussion of these data in the results and discussion sections.

- I can't quite follow the logic why the authors performed transcriptomic analysis of the SUN1 and how they chose their time points. Their data up to this point indicate that SUN1 has a structural or coordinating role in the bipartite MTOC during male gametogenesis. Based on that it is rather unlikely that SUN1 KO directly leads to transcriptional changes within the 8 min of exflagellation. Isn't it more likely that transcriptional differences are purely a downstream effect of incomplete/failed gametogenesis? This is particularly true for the comparison at 30 min, which compares a mixture of exflagellated/emerged gametes and zygotes in WT to a mixture of aberrant, arrested gametes in the knockout, which will likely not give any meaningful insight. The by far most significant GO-term is then also nuclear-transcribed mRNA catabolic process, which is likely not related at all to SUN1 function (and the authors do not even comment on this in the main text). I would therefore suggest removing the 30 min data set from this manuscript. As a minor point, I would suggest highlighting some of the top de-regulated gene IDs in the volcano plots and stating their function. Also, please state how you prepared the cells for the transcriptomes and in how many replicates this was done.

As suggested by the reviewer we have removed the 30 min post activation data from the manuscript. We have also moved the rest of the transcriptomics data to supplementary information (Fig. S5) and toned down the presentation of this aspect of the work in the results and discussion sections.

- Live-cell imaging of SUN1-GFP does nicely visualise the NE during gametogenesis, showing a highly dynamic NE forming loops and folds, which is very exciting to see. It would be beneficial to also show a video from the life-cell imaging.

We have now added videos to the manuscript as suggested by the reviewer. Please see the supplementary Videos S1 and S2.

In their discussion, the authors state multiple times that NE dynamics are changed upon SUN1 KO. Yet, they do not provide data supporting this claim, i.e. that the extended loops and folds found in the nuclear envelope during gametogenesis are affected in any way by the knockout of SUN1 or ALLAN. What happens to the NE in absence of SUN1? Are there less loops and folds? In absence of a reliable NE marker this may not be entirely easy to address, but at least some SBF-SEM images of the sun1-KO gametocytes could provide insight.

It was difficult to provide SBF-SEM images as that work is beyond the scope of this manuscript. We will consider this approach in our future work. We re-examined many of our TEM images of SUN1-KO and ALLAN-KO parasites and did find some micrographs showing aberrant nuclear membrane folding (<5%) (Please see Author response image 2). However, we also observed similar structures in some of the WT-GFP samples (<5%), so we do not think this is a strong phenotype of the SUN1 or ALLAN mutants.

**Author response image 2. sa3fig2:** 

- I think the exciting part of the manuscript is the cell biological role of SUN1 on male gametogenesis, which could be carved out a bit more by a more detailed phenotyping. Specifically it would be good to quantify(1) If DNA replication to an octoploid state still occurs in SUN1-KO and ALLAN-KO,

DNA replication is not affected in the SUN1-KO and ALLAN-KO mutants: DNA content increases to 8N (data added in Fig. 3J and Fig. S10F).

(2) The proportion of anucleated gametes in WT and the KO lines

We have added these data in Fig. 3K and Fig. S10G

(3) A quantification of the BB clustering phenotype (in which proportion of cells do the authors see this phenotype). This could be addressed by simple fixed immunofluorescence images of the respective WT/KO lines at various time points after activation (or possibly by reanalysis of the already obtained images) and would really improve the manuscript.

We have reanalysed the BB clustering phenotype and added the quantitative data in Fig. 4E and Fig. S7.

Especially the claim that emerged SUN1-KO gametes lack a nucleus is currently only based on single slices of few TEM cells and would benefit from a more thorough quantification in both SUN1- and ALLAN-Kos

We have examined many microgametes (100+ sections). In WT parasites a small proportion of gametes can appear to lack a nucleus if it does not extend all the way to the apical and basal ends (Hair et al. 2022). However, the proportion of microgametes that appear to lack a nucleus (no nucleus seen in any section) was much higher in the SUN1 mutant. In contrast, this difference was not as clear cut in the ALLAN mutant with a small proportion of intact (with axoneme and nucleus) microgametes being observed.

We have done additional analysis of male gametes, looking for the presence of the nucleus by live cell imaging after DNA staining with Hoechst. These data are added in Fig. 3K (for Sun1-KO) and Fig. S10G (for Allan-KO).

- The TEM suggests that in the SUN1-KO, kinetochores are free in the nucleus. Are all kinetochores free or do some still associate to a (minor/incorrectly formed) spindle? The authors could address this by tagging NDC80 in the KO lines.

Our observation and quantification of the data indicated that 100% of kinetochores were attached to spindle microtubules and that 0% were unattached kinetochores in the WT parasites. However, the exact opposite was found for the SUN1 mutant with 100% unattached kinetochores and 0% attached. The result was not quite as clear cut in the ALLAN mutant, with 98% unattached and 2% attached. An important observation was the lack of separation of the nuclear poles and any spindle formation. Spindle formation was never or very rarely observed in the mutants.

- Finally, I think it is curious that in contrast to SUN1, ALLAN seems to be less important, with some KO parasite completing the life cycle. Maybe a more detailed phenotyping as above gives some more hints to where the phenotypic difference between the two proteins lies. I would assume some ALLAN-KO cells can still segregate the basal body. Can the authors speculate/discuss in more detail why these two proteins seems to have slightly different phenotypes?

We agree with the reviewer. Overall, the ALLAN-KO has a less prominent phenotype than that of the Sun1-KO. The main difference is that in the ALLAN-KO mutant some basal body segregation can occur, leading to the production of some fertile microgametocytes, and ookinetes, and oocyst formation (Fig. 8). Approximately 5% of oocysts sporulated to release infective sporozoites that could infect mice in bite back experiments and complete the life cycle. In contrast the Sun1-KO mutant made no healthy oocysts, or infective sporozoites, and could not complete the life cycle in bite back experiments. We have analysed the phenotype in detail and provide quantitative data for gametocyte stages by EM and ExM in Figs. 4 and S8 (SUN1) and Figs. 7 and S11 (ALLAN). We have also performed detailed analysis of oocyst and sporozoite stages and included the data in Fig. 3 (SUN1) and S10 (ALLAN).

Based on the location, and functional and interactome data, we think that SUN1 plays a central role in coordinating nucleoplasm and cytoplasmic events as a key component of the nuclear membrane lumen, whereas ALLAN is located in the nucleoplasm. Deleting the SUN1 gene may disrupt the connection between INM and ONM whereas the deletion of ALLAN may affect only the INM.

Some additional points where the data is not entirely sound yet or could be improved:- Localisation of SUN1: There seems to be a discrepancy between SUN1-GFP location as observed by live cell microscopy, and by Expansion Microscopy (ExM), similar for ALLAN-GFP. By live-cell microscopy, the SUN1 localisation is much more evenly distributed around the NE, while the localisation in ExM is much more punctuated, and e.g. in Figure 1E seems to be within the nucleus. Do the authors have an explanation for this? Also, in Fig. 1D there are two GFP foci at the cell periphery (bottom left of the image), which I would think are not SUN1-Foci, as they seem to be outside of the cell. Is the antibody specific? Was there a negative control done for the antibody (WT cells stained with GFP antibodies after ExM)?

High resolution SIM and expansion microscopy showed that the SUN1-GFP molecules coalesce to form puncta, in contrast to the more uniform distribution observed by live cell imaging. This apparent difference may be due to a better resolution that could not be achieved by live cell imaging. We agree with the reviewer that the two green foci are outside of the cell. As a negative control we have used WT-ANKA cells (which contain no GFP) and the anti-GFP antibody, which gave no signal. This confirms the specificity of the antibody (please see the new Fig. S3).

- The authors argue that SIM gave unexpected results due to PFA fixation leading to collapse of the NE loops. However, they also fix their ExM cells and their EM cells with PFA and do not observe a collapse, at least from what I see in the two presented images and in the 3D reconstruction. Is there something else different in the sample preparation?

There was no difference in the fixation process for samples examined by SIM and ExM, but we used an anti-GFP antibody in ExM to visualise the SUN1-GFP, while in SIM the images of GFP signal were collected directly after fixation. We used both PFA and methanol as fixative, and both methods showed a coalescing of the SUN1-GFP signal (please see the new Fig. S2 and S3).

Can the authors trace their NE in ExM according to the NHS-Ester signal?

We could trace the NE in the ExM by the NHS-ester signal and observed that the SUN1-GFP signal was largely coincident with the NE (Please see the new Fig. S3B).

- Fig 2D: It would be good to not just show images of oocysts but actually quantify their size from images. Also, have the authors determined the sporozoite numbers in SUN1-KO?

We have measured oocyst size (data added in new Fig. 3) and added the sporozoite quantification data in Fig. 3D.

- Line 481-483: the authors state that oocyst size is reduced in ALLAN-KO but do not show the data. Please quantify oocyst size or at least show representative images. Also the drastic decrease in sporozoite numbers (Fig. 6D, E) is not mentioned in the text. Please add reference to Fig S7D when talking about the bite back data.

We have added the oocyst size data in Fig. S10. We mention the changes in sporozoite numbers (now shown in Fig. 7D, E), and refer to the bite back data shown in current Fig. 7E.

- Fig S1C, 6C: Both WB images are stitched, but this is not clearly indicated e.g. by leaving a small gap between the lanes. Also please show a loading control along with the western blots. Also there seems to be a (unspecific?) band in the control, running at the same height as Allan-GFP WB. What exactly is the control?

We have provided the original blot showing the bands of ALLAN-GFP and SUN1-GFP. As a positive control, we used an RNA associated protein (RAP-GFP) that is highly expressed in Plasmodium and regularly used in our lab for this purpose.

- Regarding the crossing experiment: The authors conclude from this cross that SUN1 is only needed in males, yet for this conclusion they would need to also show that a cross with a female line does not rescue the phenotype. The authors should repeat the cross with a male-deficient line to really test if the phenotype is an exclusively male phenotype. In addition, line 270-272 states that no oocysts/sporozoites were detected in sun1-ko and nek4-ko parasites. However, the figure 2E shows only oocysts, not sporozoites, and shows also that sun1-ko does form oocysts, albeit dead ones.

We have now performed the experiment of crossing the Sun1-KO parasite line with a male deficient line (Hap2-KO) and added the data in Fig. 3I. We have added images showing sporozoites in oocysts.

- In Fig S1 the authors show that they also generated a SUN1-mCherry line, yet they do not use it in any of the presented experiments (unless I missed it). Would it be beneficial to cross the SUN1-mCherry line with the Allan1-GFP line to test colocalisation (possibly also by expansion microscopy)?

We did generate a SUN1-mCherry line, with the intent to cross ALLAN-GFP and SUN1-mCherry lines and observe the co-location of the proteins. Despite multiple attempts this cross was unsuccessful. This may have been due to their close proximity such that the addition of both GFP and mCherry was difficult to facilitate a proper protein-protein interaction between either of the proteins.

- Line 498: "In a significant proportion of cells" - What was the proportion of cells, and what does significant mean in this context?

Approximately 67% of cells showed the clumping of BBs. We have now added the numbers in Figs. 6H and S11I.

- The authors should discuss a bit more how their work relates to the work of Sayers et al. 2024, which also identified the SUN1-ALLAN complex. The paper is cited, but only very briefly commented on.

We have extended this discussion now in lines 590-605.

Suggestions how to improve the writing and data presentation.- General presentation of microscopy images: Considering that large parts of the manuscript are based on microscopy data, their presentation could be improved. Single-channel microscopy images would benefit from being depicted in gray scale instead of color, which would make it easier to see the structures and intensities (especially for blue channels).

Whilst we agree with the reviewer, sometimes it is difficult to see the features in the merged images. Therefore, we would like to request to be allowed to retain the colours, which can be easily followed in both individual and merged images.

Also, it would be good to harmonize in which panels arrows are shown (e.g. Fig 1G, where some white arrows are in the SUN1-GFP panel, while others are in the merge panel, but they presumably indicate the same thing.). At the same time, Fig 1H doesn't have any with arrows, even though the figure legend states so.

We apologise for this lack of consistency, and we have now added arrows wherever they are missing to harmonise in the presentations.

Fig 3A and S4 show the same experiment but are coloured in different colours (NHS-Eester in green vs grey scale).- Are the scale bars of all expansion microscopy images adjusted for the expansion factor?

Yes, the scale bars are adjusted accordingly.

- The figure legends would benefit from streamlining, as they have very different style between figures (eg Fig. 6 which has a concise figure legend vs microscopy figures where figure legends are very long and describe not only the figure but the results)

The figure legends have been streamlined, with removal of the description of results.

- Line 155-156: The text makes it sound like the expression only happens after activation. is that the case? Are these images activated or non-activated gametocytes?

They are expressed before activation, but the signal intensifies after activation. Images from before and after activation of gametocytes have been added in Fig. S1F.

- Line 267: Reference to the original nek4-KO paper missing

This reference is now included.

- Line 301: The reference to Figure 2J seems to be a bit arbitrarily placed. Also, this schematic of lipid metabolism is never discussed in relation to the transcriptomic or lipidomic data.

We have moved these data to supplementary information and modified the text.

- Line 347-349 states that gametes emerged, but the referenced figure shows activated gametocytes before exflagellation.

We have corrected the text to the start of exflagellation.

- Line 588: Spelling mistake in SUN1-domain

Corrected.

- Line 726/731: i missing in anti-GFP

Corrected.

- Line 787-789: statement of scale bar and number of cells imaged is not at the right position in the figure legend.

Moved to right place

- Line 779, 783: "shades of green" should be just "green". Same goes for line 986, 989 with "shades of grey"

Changed.

- Line 974, 976: please correct to WT-GFP and dsun1

Corrected.

- Line 1041, 1044: WT-GFP instead of WTGFP.

Corrected to WT-GFP.

- Fig 1B, D, E, Fig S1G, H: What are the time points of imaging?

We have added the time points to the images in these figures.

- Fig 1D/Line 727: the scale of the scale bar on the inset is missing.

We have added the scale bar.

- Fig 3 E-G and 6H-J: Please indicate total number of cells/images analysed per quantification, either in the graphs themselves or in the figure legend.

We indicate now the number of cells analysed in individual figures and also in Fig. S5C and S8C, respectively.

- Fig 5B: What is NP

Nuclear Pole (NP), also known as the nuclear/acentriolar MTOC (Zeeshan et al 2022; PMID: 35550346).

- Fig S1B/D: The legend states that there is an arrow indicating the band, but there is none.

We have added the arrow.

- Fig S2C: Is the scale bar really the same for the zygote and the ookinete?

We have checked this and used the same for both zygote and ookinete.

- Fig S3C, S7C: which stages was qRT-PCR done on?

Gametocytes activated for 8 min.

- Fig. S3D, S7D: According to the figure legend, three independent experiments were performed. How many mice were used per experiment? It would be good to depict the individual data points instead of the bar graph. For S7D, 3 data points are depicted (one in WT, two in allan-KO), what do they mean?

The bite back experiment was performed using 15-20 mosquitoes infected with WT-GFP and gene knockout lines to feed on one naïve mouse each, in three different experiments. We have now included the data points in the bar diagrams.

- Fig S3: Panel letters E and G are missing

We have updated the lettering in current Fig. S5

- Fig 3D: Please indicate what those boxes are. I presume that these are the insets show in b, e and j, but it is never mentioned. J is not even larger than i. Also, f is quite cropped, it would be good to see the large-scale image it comes from to see where in the nucleus these kinetochores are placed. Were there unbound kinetochores found in WT?

We mention the boxes in the figure legends. It is rare to find unbound kinetochores in WT parasite. We provide large scale and zoomed-in images of free kinetochores in Fig. S8.

- Fig S4: Insets are not mentioned in the figure legend. Please add scale bar to zoom-ins

We now describe the insets in the figure legends and have added scale bars to the zoomed-in images.

- Fig S5A, B: Please indicate which inset belongs to which sub-panel. Where does Ac stem from?

We have now included the full image showing the inset (new Fig. S8).

- Fig S5C and S8C: Change "DNA" to "Nucleus".

We have changed “DNA” to “Nucleus”. Now they are Fig. S8K and S11I.

**Reviewer #3 (Significance):**
Yet, the statement that SUN1 is also important for lipid homoeostasis and NE dynamics is currently not backed up by sufficient data. I believe that the manuscript would benefit from removing the less convincing transcriptomic and lipidomic datasets and rather focus on more deeply characterising the cell biology of the knockouts. This way, the results would be interesting not only for parasitologists, but also for more general cell biologists.

We have moved the lipidomics and transcriptomics data to supplementary information and toned down the emphasis on these data to make the manuscript more focused on the cell biology and analysis of the genetic KO data.